# DYNAMICS-AWARE COMPARISON OF LEARNED REWARD FUNCTIONS

**Blake Wulfe, Logan Ellis, Jean Mercat, Rowan McAllister, Adrien Gaidon**
Toyota Research Institute (TRI)
{first.last}@tri.global

**Ashwin Balakrishna**
University of California, Berkeley
ashwin_balakrishna@berkely.edu

## ABSTRACT

The ability to learn reward functions plays an important role in enabling the deployment of intelligent agents in the real world. However, *comparing* reward functions, for example as a means of evaluating reward learning methods, presents a challenge. Reward functions are typically compared by considering the behavior of optimized policies, but this approach conflates deficiencies in the reward function with those of the policy search algorithm used to optimize it. To address this challenge, Gleave et al. (2020) propose the Equivalent-Policy Invariant Comparison (EPIC) distance. EPIC avoids policy optimization, but in doing so requires computing reward values at transitions that may be impossible under the system dynamics. This is problematic for learned reward functions because it entails evaluating them outside of their training distribution, resulting in inaccurate reward values that we show can render EPIC ineffective at comparing rewards. To address this problem, we propose the Dynamics-Aware Reward Distance (DARD), a new reward pseudometric. DARD uses an approximate transition model of the environment to transform reward functions into a form that allows for comparisons that are invariant to reward shaping while only evaluating reward functions on transitions close to their training distribution. Experiments in simulated physical domains demonstrate that DARD enables reliable reward comparisons without policy optimization and is significantly more predictive than baseline methods of downstream policy performance when dealing with learned reward functions.[†]

## 1 INTRODUCTION

Reward functions provide a general way to specify objectives for reinforcement learning. Tasks such as game playing have simple, natural reward functions that enable learning of effective policies (Tesauro, 1995; Mnih et al., 2013; Silver et al., 2017); however, many tasks, such as those involving human interaction, multiple competing objectives, or high-dimensional state spaces, do not have easy to define reward functions (Knox et al., 2021; Holland et al., 2013; Cabi et al., 2019). In these cases, reward functions can be learned from demonstrations, explicit preferences, or other forms of reward supervision (Ziebart et al., 2008; Christiano et al., 2017; Cabi et al., 2019). However, evaluating the quality of learned reward functions, which may be too complicated to manually inspect, remains an open challenge.

Prior work on comparing reward functions typically involves a computationally expensive and indirect process of (i) learning a reward function, (ii) learning a policy on that reward function, and (iii) evaluating that policy on a known, ground-truth reward function. Gleave et al. (2020) refer to this as the "rollout method" for evaluating methods for learning reward functions. Furthermore, the rollout method can be potentially dangerous in safety critical environments because (i) it cannot distinguish deficiencies between the reward function and the policy optimization procedure, and (ii) rewards are

---

[†]Videos available at https://sites.google.com/view/dard-paper

evaluated on the distribution of transitions induced by an optimized policy, which may not capture all the transitions required to accurately quantify reward differences.

To address these challenges, Gleave et al. (2020) introduce Equivalent-Policy Invariant Comparison (EPIC). EPIC quantifies differences in reward functions directly, without learning policies. EPIC uses an offline dataset of transitions, and consists of two stages. First, transitions from this dataset are used to convert reward functions to a canonical form that is invariant to reward transformations that do not affect the optimal policy. Second, the correlation between reward values is computed on transition samples, yielding a pseudometric capturing reward function similarity. However, when computing this canonical form, EPIC evaluates rewards on transitions between all state-state combinations, regardless of whether those state-state transitions are possible or not. While this makes it possible for EPIC to quantify reward distances that are robust to arbitrary changes in transition dynamics, in practice evaluating learned reward functions on physically impossible transitions leads to unreliable reward values, as these are outside the distribution of the transitions on which the reward functions are learned. We show empirically in Section 5 that these out-of-distribution reward evaluations can render EPIC an ineffective reward comparison metric.

To address this issue, we propose a new method for quantifying differences in reward functions, Dynamics-Aware Reward Distance (DARD), that transforms reward functions into a form allowing for reward-shaping-invariant comparisons while ensuring that learned reward functions are only evaluated on transitions that are approximately physically realizable. We achieve this by assuming access to a transition model for the environment of interest, and evaluate reward functions on transitions sampled from this transition model. As a result, DARD only evaluates learned reward functions close to their training distribution, and is well motivated in a variety of physical environments in which transition models are unlikely to change significantly between learning and deployment. Experiments suggest that DARD better reflects reward function similarity than does EPIC in comparing learned reward functions in two simulated, physics-based environments: the first involving multi-agent navigation, and the second involving manipulating a robotic arm.

## 2 RELATED WORK

### 2.1 LEARNING REWARD FUNCTIONS

A variety of methods exist for learning reward functions, which can be categorized based on the form of supervision used. Rewards can be learned from supervision provided by explicit labels. For example, human-labeled preferences between demonstrations can be used to learn reward functions (Christiano et al., 2017). Rewards can also be labeled on a per-timestep basis, allowing for learning via supervised regression (Cabi et al., 2019).

Alternatively, inverse reinforcement learning (IRL) methods typically assume access to expert demonstrations, and attempt to find a reward function under which the expert is uniquely optimal (Ziebart et al., 2008). IRL is underconstrained in that many reward functions can make the expert policy appear optimal. This problem is often addressed by favoring reward functions that induce high-entropy optimal policies (Ziebart et al., 2008). Recent work in IRL focuses on learning reward functions with expressive function approximators such as neural networks, particularly in an adversarial formulation (Fu et al., 2017). There has also been recent interest in offline IRL methods (Lee et al., 2019). In this work, we study how these various reward learning methods can be evaluated efficiently and accurately in practice.

### 2.2 EVALUATING METHODS OF LEARNING REWARD FUNCTIONS

Reward functions are typically evaluated by first learning an optimal policy under the reward function. The value of the induced policy is found by aggregating the returns from online trajectory rollouts under the known, ground-truth reward function. This requirement of online rollouts from experimental policies is often prohibitive in safety-critical domains such as autonomous driving or decision making in healthcare (Thapa et al., 2005).

Off-policy evaluation (OPE) methods attempt to address these issues by estimating the value of a policy in a purely offline setting by using previously collected data. Traditional OPE methods address policy mismatch by using a form of importance sampling (Precup et al., 2000). This approach

can be effective, but is unbiased only if the underlying expert policy is known, and can be subject to large variance. Direct OPE methods (Dudik et al. (2011)), apply regression-based techniques in either a model-based (Paduraru, 2013) or model-free setting (Le et al., 2019). Contemporary works combine these two methods using Doubly Robust Estimation, which originated in a contextual bandits setting (Dudik et al., 2011), and was later extended to reinforcement learning by Jiang & Li (2016). More recent works seek to forego models and importance sampling entirely, such as Irpan et al. (2019), which recasts OPE as a positive-unlabeled classification problem with the Q-function as the decision function.

All OPE methods indirectly evaluate the reward by estimating the value of the induced optimal policy. This introduces additional complexity into the evaluation procedure and can obfuscate sources of error as resulting either from aspects of policy learning or due to the underlying reward function. EPIC (Gleave et al., 2020) seeks to circumvent these shortcomings by comparing reward functions directly. Our work builds on EPIC as covered in Section 4, but uses a transition model of the environment to significantly improve the accuracy of comparisons involving learned reward functions.

## 3 PRELIMINARIES

### 3.1 MARKOV DECISION PROCESSES

A Markov decision process (MDP) is a tuple $(\mathcal{S}, \mathcal{A}, T, R, \gamma, d_0)$ where $\mathcal{S}$ and $\mathcal{A}$ are the state and action spaces. The transition model $T : \mathcal{S} \times \mathcal{A} \times \mathcal{S} \rightarrow [0, 1]$ maps a state and action to a probability distribution over next states, the reward $R : \mathcal{S} \times \mathcal{A} \times \mathcal{S} \rightarrow \mathbb{R}$ measures the quality of a transition. Finally, $\gamma \in [0, 1]$ and $d_0$ are respectively the discount factor and initial state distribution. A trajectory $\tau$ consists of a sequence of state-action pairs: $\tau = \{(s_0, a_0), (s_1, a_1), ...\}$, and the return $g$ of a trajectory is defined as the sum of discounted rewards along that trajectory: $g(\tau) = \sum_{t=0}^{\infty} \gamma^t R(s_t, a_t, s_{t+1})$. The goal in an MDP is to find a policy $\pi : \mathcal{S} \times \mathcal{A} \rightarrow [0, 1]$ that maximizes the expected return, $\mathbb{E}[g(\tau)]$, where $s_0 \sim d_0$, $a_t \sim \pi(a_t|s_t)$, and $s_{t+1} \sim T(s_{t+1}|s_t, a_t)$.

We denote a distribution over a space $\mathcal{S}$ as $\mathcal{D}_{\mathcal{S}}$, and the set of distributions over a space $\mathcal{S}$ as $\Delta(\mathcal{S})$.

### 3.2 REWARD FUNCTION EQUIVALENCE

Our goal is to define a metric for comparing reward functions without learning policies. Nevertheless, the measure of similarity we are interested in is the extent to which the optimal policies induced by two reward functions are the same. Because different reward functions can induce the same set of optimal policies, we in fact seek a *pseudometric* as formalized in the following definition.

**Definition 3.1** (Pseudometric (Definition 3.5 from Gleave et al. (2020))). *Let $\mathcal{X}$ be a set and $d : \mathcal{X} \times \mathcal{X} \rightarrow [0, \infty]$ a function. $d$ is a premetric if $d(x, x) = 0$ for all $x \in \mathcal{X}$. $d$ is a pseudometric if, furthermore, it is symmetric, $d(x, y) = d(y, x)$ for all $x, y \in \mathcal{X}$; and satisfies the triangle inequality, $d(x, z) \leq d(x, y) + d(y, z)$ for all $x, y, z \in \mathcal{X}$. $d$ is a metric if, furthermore, for all $x, y \in \mathcal{X}$, $d(x, y) = 0 \implies x = y$.*

Which rewards induce the same set of optimal policies? Ng et al. (1999) showed that without additional prior knowledge about the MDP only reward functions that are related through a difference in state potentials are equivalent. This "reward shaping" refers to an additive transformation $F$ applied to an initial reward function $R$ to compute a shaped reward $R'$: $R'(s, a, s') = R(s, a, s') + F(s, a, s')$. Reward shaping is typically applied in a reward design setting to produce denser reward functions that make policy optimization easier in practice (Ng et al., 2003; Schulman et al., 2015). Ng et al. (1999) showed that if $F$ is of the form $F(s, a, s') = \gamma \Phi(s') - \Phi(s)$ for arbitrary state potential function $\Phi(s)$, the set of optimal policies under $R$ and $R'$ are the same. This provides a definition for reward equivalence:

**Definition 3.2** (Reward Equivalence, Definition 3.3 from Gleave et al. (2020)). *Bounded rewards $R_A$ and $R_B$, are equivalent if and only if*

$$\exists \lambda > 0, \text{ and } \Phi : \mathcal{S} \rightarrow \mathbb{R} \text{ bounded s.t.}$$

$$R_B(s, a, s') = \lambda R_A(s, a, s') + \gamma \Phi(s') - \Phi(s) \; \forall s, s' \in \mathcal{S}, \; a \in \mathcal{A}.$$

This form of equivalence is only possible in reward functions that depend on $(s, a, s')$, which motivates our focus on these types of rewards in this paper. This choice is further motivated by the fact that there are cases where conditioning on the next state $s'$ can make reward learning simpler, for example in highly stochastic MDPs, or in cases where rewards are a simple function of $(s, a, s')$, but not of $(s, a)$. We next describe an existing pseudometric that in theory is capable of identifying these reward equivalence classes.

### 3.3 Equivalent-Policy Invariant Comparison (EPIC)

EPIC is a pseudometric for comparing reward functions and involves two stages. The first stage converts reward functions into a canonical form that removes the effect of reward shaping, and the second stage computes the correlation between the canonicalized rewards.

The following definition describes the canonicalization stage.

**Definition 3.3** (Canonically Shaped Reward (Definition 4.1 from Gleave et al. (2020))). *Let $R : \mathcal{S} \times \mathcal{A} \times \mathcal{S} \to \mathbb{R}$ be a reward function. Given distributions $\mathcal{D}_\mathcal{S} \in \Delta(\mathcal{S})$ and $\mathcal{D}_\mathcal{A} \in \Delta(\mathcal{A})$ over states and actions, let $S$ and $S'$ be independent random variables distributed as $\mathcal{D}_\mathcal{S}$ and $A$ distributed as $\mathcal{D}_\mathcal{A}$. Define the canonically shaped $R$ to be:*

$$C_{\mathcal{D}_\mathcal{S}, \mathcal{D}_\mathcal{A}}(R)(s, a, s') = R(s, a, s') + \mathbb{E}[\gamma R(s', A, S') - R(s, A, S') - \gamma R(S, A, S')]. \quad (1)$$

Fig. 1 visualizes this process. Informally, this operation computes the mean of the potentials of $s$ and $s'$ with respect to $S'$ using only evaluations of the reward $R$, and subtracts/adds those averages from/to the original, shaped reward, thereby canceling out their effects. This cancellation behavior is formalized in the following proposition.

**Proposition 1** (The Canonically Shaped Reward is Invariant to Shaping (Proposition 4.2 from Gleave et al. (2020))). *Let $R : \mathcal{S} \times \mathcal{A} \times \mathcal{S} \to \mathbb{R}$ be a reward function, and $\Phi : \mathcal{S} \to \mathbb{R}$ a potential function. Let $\gamma \in [0, 1]$ be a discount rate, and $\mathcal{D}_\mathcal{S} \in \Delta(\mathcal{S})$ and $\mathcal{D}_\mathcal{A} \in \Delta(\mathcal{A})$ be distributions over states and actions. Let $R'$ denote $R$ shaped by $\Phi : R'(s, a, s') = R(s, a, s') + \gamma\Phi(s') - \Phi(s)$. Then the canonically shaped $R'$ and $R$ are equal: $C_{\mathcal{D}_\mathcal{S}, \mathcal{D}_\mathcal{A}}(R') = C_{\mathcal{D}_\mathcal{S}, \mathcal{D}_\mathcal{A}}(R)$.*

This invariance means that canonicalized rewards can be compared without reward shaping impacting the comparison. In practice, canonicalization is performed over an offline dataset of transitions. The rewards for $(s, a, s')$ transitions are computed, and then canonicalized using (1). The distributions $\mathcal{D}_\mathcal{S}$ and $\mathcal{D}_\mathcal{A}$ are the marginal distributions of states and actions in the dataset on which evaluation proceeds. As we will discuss in the next section, this results in evaluating learned reward functions outside their training distribution, leading to inaccurate reward values.

EPIC's second stage computes the Pearson distance between two canonicalized reward functions.

**Definition 3.4** (Pearson Distance (Definition 4.4 from Gleave et al. (2020))). *The Pearson distance between random variables $X$ and $Y$ is defined by the expression $D_\rho(X, Y) = \sqrt{1 - \rho(X, Y)}/\sqrt{(2)}$, where $\rho(X, Y)$ is the Pearson correlation between X and Y.*

In combining these two steps, EPIC is capable of quantifying the similarity of two reward functions while being invariant to shaping (due to canonicalization), and invariant to changes in shift and scale (due to the use of the Pearson distance).

**Definition 3.5** (EPIC pseudometric (Definition 4.6 from Gleave et al. (2020))). *Let $S$, $A$, $S'$ be random variables jointly following some coverage transition distribution. Let $\mathcal{D}_\mathcal{S}$, and $\mathcal{D}_\mathcal{A}$ respectively be the state and action distributions. The EPIC pseudometric between rewards $R_A$ and $R_B$ is:*

$$D_{EPIC}(R_A, R_B) = D_\rho\Big(C_{\mathcal{D}_\mathcal{S}, \mathcal{D}_\mathcal{A}}(R_A)(S, A, S'), C_{\mathcal{D}_\mathcal{S}, \mathcal{D}_\mathcal{A}}(R_B)(S, A, S')\Big). \quad (2)$$

Note that during the canonicalization stage, $S$, $A$, $S'$ are independent while in the stage computing the Pearson correlation between canonical rewards, the joint distribution of $S$, $A$, $S'$ is used.

## 4 DARD: Dynamics-Aware Reward Distance

Here we present DARD, a new distance measure between learned reward functions that uses information about system dynamics to prevent errors due to out-of-distribution queries of learned

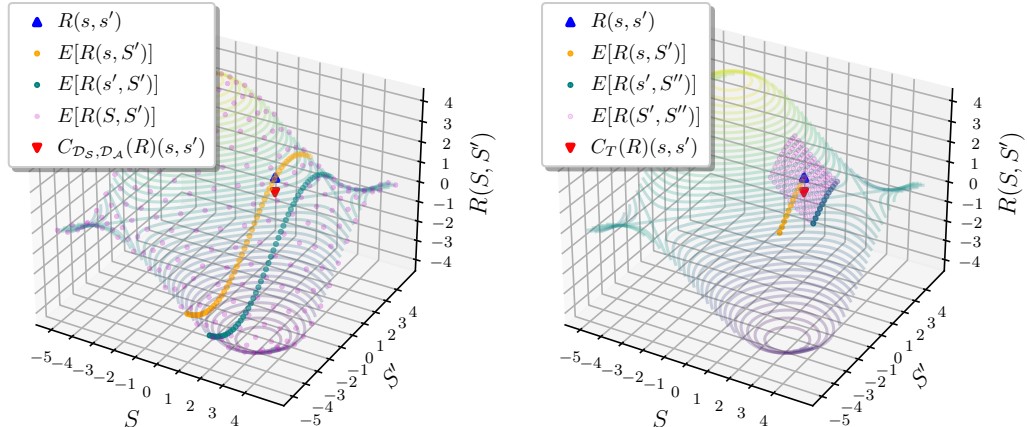

Figure 1: These plots depict the canonicalization procedure of EPIC (**left**) and the transformation of DARD (**right**) applied to a single transition from $s = 1$ to $s' = 2$ in a simple MDP. The contours depict a reward function for a 1D, continuous-state environment with a continuous action space of displacements in $[-1, 1]$. The axes show the state $S$ and next state $S'$ (omitting the action because $T$ is deterministic). The reward function is zero everywhere, but shaped by the potential function $\Phi(s)$. The plotted points correspond to different terms in Eq. (1) or Eq. (3) as indicated by the legends. EPIC canonicalizes reward functions by computing reward values for transitions that may not be possible under the transition model of the environment. This is reflected in the visualization by samples comprising the different expectations being dispersed globally over $S$ and/or $S'$. Our approach in contrast uses a transition model to transform the reward function using approximately in-distribution transitions as illustrated visually by its local sampling in computing the expectations. Both approaches arrive at the unshaped value of zero depicted by the red triangle.

rewards. As described in 3.3, the EPIC distance measure first transforms reward functions as shown in equation (1). This transformation procedure involves computing expectations $\mathbb{E}[R(s', A, S')]$, $\mathbb{E}[R(s, A, S')]$, and $\mathbb{E}[R(S, A, S')]$ with *independent* random variables $S$ and $S'$ distributed as $\mathcal{D}_S$, which in practice is the marginal distribution of states in the dataset. Thus, the resulting transitions may not actually be possible under the transition model of the environment of interest. Computing transformed rewards with these out-of-distribution transitions allows EPIC to be robust to (arbitrary) changes in the transition model. Moreover, without additional information about the transition model, two reward functions that behave identically on in-distribution transitions, but differently on out-of-distribution transitions, could justifiably be regarded as different.

However, if these reward functions are learned from data, evaluating them on out-of-distribution transitions can lead to unreliable reward values, introducing errors into the reward distance computed by EPIC. For example, the reward value of a transition in which a home robot teleports across a room is likely to be arbitrarily incorrect, since no similar transition would be observed when the robot interacts with its environment. Because EPIC necessarily evaluates learned reward functions on out-of-distribution data when computing reward distances, these distances can have significant inaccuracies as we show empirically in Section 5.

To address this challenge, we assume access to a transition model of the environment (which may also be learned in practice), and propose Dynamics-Aware Reward Distance (DARD), a new reward distance measure, which builds on EPIC. DARD computes an alternative reward transformation that evaluates reward functions on transitions that are much closer to those observed in the environment. This helps to encourage evaluation of reward functions closer to the distribution of transitions on which they were trained while still providing a distance metric between reward functions that is invariant to potential shaping.

**Definition 4.1** (Dynamics-Aware Transformation). *Let $R : \mathcal{S} \times \mathcal{A} \times \mathcal{S} \to \mathbb{R}$ be a reward function. Given distributions $\mathcal{D}_S \in \Delta(\mathcal{S})$ and $\mathcal{D}_A \in \Delta(\mathcal{A})$ over states and actions, let $S$ be a random variable distributed as $\mathcal{D}_S$, and $A$ be a random variable distributed as $\mathcal{D}_A$. Furthermore, given a probabilistic transition model defining a conditional distribution over next states $T(S'|S, A)$, let*

$S'$ and $S''$ be random variables distributed as $T(S''|s, A)$ and $T(S''|s', A)$ respectively. Define the dynamics-aware transformation of $R$ as follows:

$$C_T(R)(s, a, s') = R(s, a, s') + \mathbb{E}[\gamma R(s', A, S'') - R(s, A, S') - \gamma R(S', A, S'')]. \qquad (3)$$

By sampling next states from the transition model conditioning on the current state (either $s$ or $s'$), this transformation is able to evaluate reward models at $(s, a, s')$ transitions that are closer to their training distribution. In the case of the term $\mathbb{E}[R(S', A, S'')]$, the transitions evaluated are not perfectly in-distribution with respect to $T$. This is because $S''$ is distributed conditionally on the original $s'$ as opposed to the random variable $S'$. These transitions are nevertheless much closer to physically realizable than those sampled in EPIC. In Section 5 we show that learned reward functions are relatively insensitive to these slight errors in practice. We select $\mathcal{D}_{\mathcal{A}}$ to densely cover the space of possible actions. We do this because during policy optimization, policies may select a wide range of possible actions, resulting in evaluating learned reward functions on this same distribution of actions. See Figure 1 for a visualization of our dynamics-aware transformation. Informally, this operation normalizes the reward values of a transition $(s, a, s')$ based on the reward values of nearby transitions sampled from the transition model. This normalization allows for a comparison between reward functions that is invariant to potential shaping.

**Proposition 2** (The dynamics-aware transformed reward is invariant to shaping). *Let $R : \mathcal{S} \times \mathcal{A} \times \mathcal{S} \to \mathbb{R}$ be a reward function, and $\Phi : \mathcal{S} \to \mathbb{R}$ a state potential function. Let $\gamma \in [0, 1]$ be a discount factor, and $\mathcal{D}_{\mathcal{S}} \in \Delta(\mathcal{S})$ and $\mathcal{D}_{\mathcal{A}} \in \Delta(\mathcal{A})$ be distributions over states and actions, and $T : \mathcal{S} \times \mathcal{A} \to \Delta(\mathcal{S})$ be a conditional distribution over next states. Let $R'$ denote $R$ shaped by $\Phi : R'(s, a, s') = R(s, a, s') + \gamma\Phi(s') - \Phi(s)$. Then the transition-relative transformations of $R'$ and $R$ are equal: $C_T(R') = C_T(R)$.*

*Proof:* See Section A.4.

**Definition 4.2** (DARD Distance Measure). *Let $S$, $A$, $S'$ be random variables jointly following some coverage transition distribution, and let $T$ be the transition model of the corresponding environment. We define the DARD pseudometric between rewards $R_A$ and $R_B$ as follows:*

$$D_{DARD}(R_A, R_B) = D_\rho\Big(C_T(R_A)(S, A, S'), C_T(R_B)(S, A, S')\Big). \qquad (4)$$

We prove that the above distance measure is a pseudometric in Section A.4.

The key insight behind DARD is that we can use knowledge of transition dynamics to define a distance measure between reward functions that only evaluates those functions at realizable transitions. Though, unlike EPIC, DARD assumes knowledge of transition dynamics and that these dynamics will remain similar during deployment, DARD uses these additional assumptions to produce significantly more accurate and predictive distances between learned reward functions in practice.

## 5 EXPERIMENTS

In experiments we evaluate the following hypotheses: **H1**: EPIC is a poor predictor of policy return for learned reward functions due to its out-of-distribution evaluation of these reward functions. **H2**: DARD is predictive of policy return even for learned reward models, while maintaining EPIC's invariance to reward shaping. **H3**: DARD is predictive of policy performance across diverse coverage distributions. **H4**: DARD is predictive of policy return when using learned, *high-fidelity* transition models. We test these hypotheses by computing each metric across a set of reward models, and then evaluate a policy trained on each reward model against a ground truth reward function through on-policy evaluation. We perform these steps across two environments as described next.

### 5.1 EXPERIMENTAL SETUP

**Environments**  We evaluate DARD in two environments. We first study the Bouncing Balls environment, which requires an agent to navigate from a random initial position to a random goal position while avoiding a set of balls randomly bouncing around the scene. This environment captures elements of real-world, physical environments with multiple interacting agents, as is the case, for example, in autonomous driving. In these types of environments, EPIC samples transitions in a

manner such that agents teleport around the scene, causing issues for learned reward models. The second environment is adapted from the Reacher environment from OpenAI Gym (Brockman et al., 2016). This environment captures aspects of robotic manipulation tasks where arbitrary changes in joint configuration between $s$ and $s'$ can cause issues for learned reward models. Videos illustrating both environments can be found at `https://sites.google.com/view/dard-paper`.

**Coverage Distributions** In order to compute reward distances for DARD, we must sample transitions from the environments. In the experiments we consider two coverage distributions. The first is the distribution induced by a policy that takes actions uniformly at random across magnitude bounds (denoted by $\pi_{\text{uni}}$). The second is the distribution over transitions of an expert policy $\pi^*$, which we learn in both environments using Proximal Policy Optimization (PPO) Schulman et al. (2017).

**Action Sampling and Transition Models** Both environments have a 2D, continuous action space. DARD samples actions from these spaces (i.e., defines $\mathcal{D}_{\mathcal{A}}$) by defining an uniformly-spaced, 2D grid of action values within magnitude bounds.

We consider two versions of DARD in the experiments. The first uses (approximate) ground-truth transition models, and the second uses learned transition models, which we refer to as DARD-L in the results. For the ground-truth transition model in the Bouncing Balls environment we use a constant velocity model that deterministically samples next states conditional on each sampled action. For the Reacher environment, we use the underlying MuJoCo simulator (Todorov et al., 2012) as the transition model, again deterministically sampling next states.

We additionally learn neural network transition models for each environment. In the Bouncing Balls environment, we learn a transition model that independently predicts position deltas for each ball based on its physical state and the action (in the case of the controlled agent). In the Reacher environment, we learn a transition model that takes as input the state and action, and predicts a state delta used to produce the next state. See Appendix A.1.4 for model architecture and training details.

**Reward Models** For each environment, we define or learn the following reward models:

GROUND TRUTH: The reward function against which we perform on-policy evaluation. In both environments, this is a goal-based reward given when the agent position is within some range of the goal location. The Reacher environment also contains additive terms penalizing distance from the goal and large action magnitudes.

SHAPED: The ground truth reward function, shaped by a potential function. In both environments, we use a potential shaping based on the distance to the goal location. In the Bouncing Balls environment, we take the difference in the square root of the distance to goal (due to the large distance values). In the Reacher environment we use the difference of distances from the end effector to goal directly.

FEASIBILITY: A manually-defined reward model that returns the shaped reward function when evaluated with a dynamically-feasible transition, and returns unit Gaussian noise otherwise. This serves as a sanity check to illustrate that EPIC performs poorly on models that exhibit inaccurate reward values on OOD transitions.

REGRESS: A learned model trained via regression to the *shaped* reward model. We regress the shaped reward function because otherwise the metrics do not need to account for shaping (see Appendix A.2.1 for an example). Although simple, this method is used in practice in cases where people manually label reward functions (Cabi et al., 2019), and was used as a baseline in Christiano et al. (2017) ("target" method, section 3.3).

REGRESS OUT-OF-DISTRIBUTION: A model trained similarly to REGRESS, except that it is trained "out-of-distribution" in the sense that during training it samples $(a, s')$ pairs randomly from the dataset such that its training distribution matches the evaluation distribution of EPIC.

PREFERENCES: A reward model learned using preference comparisons, similar to the approach from Christiano et al. (2017). We synthesize preference labels based on ground-truth return.

All learned models are fully-connected neural networks. In the Bouncing Balls environment, the input to the network is a specific set of features designed to be susceptible to OOD issues (a squared

distance is computed between $s$ and $s'$ as one of the features). For the Reacher environment, we concatenate $s$, $a$, and $s'$ as in the learned-reward experiments in Gleave et al. (2020).

**Baselines** We compare with two baseline methods. First, we compare with EPIC. Second, we compare with a simple calculation of the Pearson distance ($D_\rho$). We include this baseline in order to assess the extent to which reward shaping is an issue that needs to be accounted for in comparing rewards (see Appendix A.2.1 for additional motivation).

## 5.2 RESULTS AND DISCUSSION

Table 1: Results for the Bouncing Balls (**Top**) and Reacher (**Bottom**) environments. **Center**: For each reward model, we compute each distance metric×1000 from GROUND TRUTH (GT). The coverage distribution is indicated by the policy ($\pi_{\text{uni}}$ corresponding to the distribution resulting from uniformly random actions, $\pi^*$ to that from expert actions). **Right**: The average episode return and its standard error of the policy trained on the corresponding reward model. Values are averaged over 5 executions of all steps (data collection, reward learning, reward evaluation) across different random seeds (see Appendix A.1.7 for standard errors of distances). Distance values that are inversely corre­lated with episode return are desirable (i.e., *increasing* distance within a column should correspond with *decreasing* episode return). Additionally, for hand-designed models that are equivalent to GT (SHAPING, FEASIBILITY), and for well-fit learned models (REGRESS), lower is better. Learned models are fit to SHAPED, and may produce higher episode return than those fit to GT as a result.

| Reward Function | 1000 x $D_{DARD}$ | | 1000 x $D_{DARD-L}$ | | 1000 x $D_{EPIC}$ | | 1000 x $D_\rho$ | | Episode Return |
|---|---|---|---|---|---|---|---|---|---|
| | $\pi_{\text{uni}}$ | $\pi^*$ | $\pi_{\text{uni}}$ | $\pi^*$ | $\pi_{\text{uni}}$ | $\pi^*$ | $\pi_{\text{uni}}$ | $\pi^*$ | |
| GT | 0.00 | 0.00 | 0.00 | 0.00 | 0.00 | 0.00 | 0.00 | 0.00 | $9.61 \pm 0.41$ |
| SHAPING | 0.00 | 0.00 | 0.00 | 0.00 | 0.00 | 0.00 | 642 | 602 | $12.0 \pm 0.20$ |
| FEASIBILITY | 0.00 | 0.00 | 0.00 | 0.00 | 642 | 601 | 642 | 602 | $12.2 \pm 0.11$ |
| REGRESS OOD | 12.3 | 9.63 | 12.3 | 9.55 | 14.4 | 6.20 | 642 | 602 | $11.9 \pm 0.13$ |
| REGRESS | 28.4 | 12.7 | 29.0 | 12.6 | 690 | 613 | 642 | 602 | $12.0 \pm 0.13$ |
| PREF | 110 | 60.0 | 108 | 58.9 | 697 | 650 | 668 | 639 | $10.8 \pm 0.13$ |

| Reward Function | 1000 x $D_{DARD}$ | | 1000 x $D_{DARD-L}$ | | 1000 x $D_{EPIC}$ | | 1000 x $D_\rho$ | | Episode Return |
|---|---|---|---|---|---|---|---|---|---|
| | $\pi_{\text{uni}}$ | $\pi^*$ | $\pi_{\text{uni}}$ | $\pi^*$ | $\pi_{\text{uni}}$ | $\pi^*$ | $\pi_{\text{uni}}$ | $\pi^*$ | |
| GT | 0.00 | 0.00 | 0.00 | 0.00 | 0.00 | 0.00 | 0.00 | 0.00 | $82.4 \pm 1.13$ |
| SHAPING | 0.00 | 0.00 | 0.00 | 0.00 | 0.00 | 0.00 | 369 | 88.9 | $78.4 \pm 3.22$ |
| FEASIBILITY | 0.00 | 0.00 | 0.00 | 0.00 | 529 | 221 | 368 | 88.3 | $83.4 \pm 0.68$ |
| REGRESS OOD | 30.9 | 130 | 29.6 | 100 | 27.3 | 60.8 | 370 | 105 | $80.4 \pm 0.86$ |
| REGRESS | 49.7 | 184 | 46.6 | 143 | 182 | 175 | 371 | 112 | $82.3 \pm 1.30$ |
| PREF | 63.5 | 251 | 63.4 | 210 | 165 | 170 | 370 | 136 | $32.8 \pm 4.99$ |

Table 1 summarizes the experimental results, which provide support for **H1** across both environ­ments. We first note as a sanity check that EPIC assigns large distances between GROUND TRUTH and FEASIBILITY. This demonstrates EPIC's tendency to assign large distances to reward functions that are equivalent when assuming a fixed transition model, but different under an arbitrary transi­tion model. This behavior manifests with learned models in practice due to EPIC's OOD evaluation of those models. For example, in both environments EPIC assigns large distances to the REGRESS model when, based on policy performance, we see that it is quite similar to the ground truth (disre­garding shaping). This results from EPIC's evaluation of these reward models with OOD transitions, which is illustrated by the fact that the metric is more much predictive of on-policy performance on the REGRESS OOD model (which does not suffer from OOD evaluation issues). Perhaps most importantly, EPIC fails to be predictive of episode return when comparing learned models. For ex­ample, in the Reacher environment, the metric assigns a lower distance to PREFERENCES (PREF) than REGRESS despite the lower episode returns of PREF.

The results from both environments also support **H2**. In the Bouncing Balls environment, DARD gives low distances for the well-fit models (FEASIBILITY, REGRESS, REGRESS OOD) as a result of only evaluating them in their training distribution. In the Reacher environment, DARD exhibits a gap between REGRESS and REGRESS OOD, which indicates that the algorithm's evaluation of the reward model on slightly out-of-distribution transitions does hurt performance. Nevertheless, on the

reward models that can be learned in practice (REGRESS, PREF) DARD produces distances that are more predictive of episode return compared to baseline methods.

The results also demonstrate that DARD is predictive of episode return across coverage distributions (**H3**). DARD is sensitive to the choice of coverage distribution because the transitions the metric samples are tightly grouped around this distribution. This is reflected in the different distance scales between $\pi_{\text{uni}}$ and $\pi^*$. Nevertheless, the relative value of DARD within a sampling distribution is still predictive of episode return. For example, the metric maintains a consistent ordering of the models across $\pi_{\text{uni}}$ and $\pi^*$ in both environments.

Finally, the results largely support the hypothesis (**H4**) that replacing ground truth transition models in DARD with *high-fidelity* learned ones still results in distance values predictive of policy return. This conclusion is supported by DARD-L's predictiveness of policy return across both environments and coverage distributions. In most cases, the distance values of DARD-L closely match those of DARD. Three reasons for this close correspondence are: (i) high-fidelity transition models can be fit in the relatively low-dimensional environments considered, (ii) DARD uses transition models to make single-step predictions, which avoids the compounding errors that may occur when sampling rollouts, and (iii) the environments and coverage distributions considered generally produce reward functions insensitive to small errors in the predicted states. The lower distance values of DARD-L in the Reacher environment when using the expert coverage distribution are an exception to this trend. By analyzing the errors made by the learned transition model, we concluded that the lower distances of DARD-L result from the reward functions being quite sensitive to small errors in predicted states in this case (see Appendix A.3.2 for detailed analysis). Although the distance values are still predictive of policy return, this result is an instance of the false positive equivalence that can result from errors in the transition model (as discussed in Appendix A.3.1 (case 2.B)), and illustrates the importance of applying DARD with well-fit learned transitions models.

**Limitations**  In our experiments, we discovered that DARD is poorly suited for reward comparison in environments where nearby transitions exclusively have similar reward values. This is the case, for example, in robotic domains with small timesteps and in which a smooth reward function is used (such as negative distance to the goal). We illustrate this behavior and explain why it occurs in Appendix A.2.1 with experiments on the Point Maze environment used by Gleave et al. (2020). In short, because DARD performs reward comparisons between transitions relative to a transition model, if all the transitions it considers (across all possible actions) have nearly-identical reward values, it can be sensitive to noise in the reward model. For this reason, DARD should be used in environments for which, at least for some parts of the state-action space, nearby transitions yield well-differentiated reward values.

Overall, these experiments demonstrate that in situations where we have confidence the transition model will not change significantly, as is the case in many physical settings, DARD allows for comparing *learned* reward functions with other reward functions without policy optimization.

## 6  CONCLUSION

The ability to reliably compare reward functions is a critical tool for ensuring that intelligent systems have objectives that are aligned with human intentions. The typical approach to comparing reward functions, which involves learning policies on them, conflates errors in the reward with errors in policy optimization. We present Dynamics-Aware Reward Distance (DARD), a pseudometric that allows for comparing rewards directly while being invariant to reward shaping. DARD uses knowledge of the transition dynamics to avoid evaluating *learned* reward functions far outside their training distribution. Experiments suggest that for environments where we can assume the transition dynamics will not change significantly between evaluation and deployment time that DARD reliably predicts policy performance associated with a reward function, even learned ones, without learning policies and without requiring online environment interaction.

There are a number of exciting areas for future work. For example, DARD could be used to define benchmarks for evaluating reward learning methods (e.g., in applications such as autonomous driving or robotics; see Appendix A.3.3). Second, DARD could be applied to high-dimensional (e.g., visual) domains where learned transition models are necessary and for which OOD evaluation issues are likely particularly challenging to overcome.

## REPRODUCIBILITY STATEMENT

We have attempted to make the experimental results easily reproducible by providing the following: first, source code necessary to reproduce results is available in the supplementary material. Second, the software for this research is designed to allow for reproducing the results in the main body of the text through a single program execution. Third, results are computed over five random seeds and standard error values are reported in Appendix A.1.7 to provide reasonable bounds within which reproduced results should fall. Fourth, we have attempted to provide extensive details regarding the experimental setup in Appendix A.1.

## ETHICS STATEMENT

The ability to reliably and efficiently evaluate reward functions without executing policies in the environment has the potential to ensure that AI systems will exhibit behavior that is aligned with the objective of the system designer before risking deploying these systems for making safety critical decisions. This has the potential to facilitate the application of reinforcement learning algorithms in settings in which this might otherwise be too risky, such as healthcare, finance, and robotics.

The primary risk associated with these methods is that system designers may rely on them inappropriately (e.g., in situations where they do not apply, or in lieu of additional validation procedures), which could result in unsafe system deployment. We have attempted to address this risk by stating limitations of the method (see Section 5.2), and by providing example applications we believe to be appropriate (see Appendix A.3.3).

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

# A    SUPPLEMENTARY MATERIAL

## A.1    EXPERIMENTAL DETAILS

The following sub-sections provide details on the different components involved in the experiments.

### A.1.1    ENVIRONMENTS

**Bouncing Balls Environment**    This is a custom environment with traits of multi-agent navigation environments such as autonomous driving. The agent attempts to navigate to a random goal location while avoiding a set of balls (other agents) bouncing around the scene. The state consists of the position and velocity of the balls (the agent included), as well as the goal location. The action space is a 2D continuous space of accelerations between $[-5, 5]$ to apply to the agent state. The transition function applies the agent accelerations to its state, while the other agents apply random accelerations sampled from a zero-mean, low variance Gaussian distribution. The reward function gives a positive value once the agent is within some distance of the goal, which results in the agent and goal position being randomly moved. This environment executes for a fixed horizon of 400 timesteps to approximate a continuing, infinite horizon MDP. We make the environment approximately continuing in order to ensure that $S$ and $S'$ are distributed identically as is required by EPIC. Sampling $S$ and $S'$ from their marginal distributions in the dataset results in transitions where the agents in $s$ have completely different positions and velocities as in $s'$, an issue that would be encountered in real-world multi-agent settings.

**Reacher Environment**    This environment is adapted from the Reacher environment from OpenAI gym (Brockman et al., 2016). We adapt the environment in two ways. First, we add a goal-based reward component. Second, we increase the timesteps between states to 5 from an original 2. We make both these changes so that the rewards of nearby transitions are sufficiently differentiated (see A.2.1 for detailed motivation).

### A.1.2    POLICY LEARNING

We used the RLlib  (Liang et al., 2018) implementation of Proximal Policy Optimization (PPO) for learning control policies (Schulman et al., 2017). Parameters are provided in table 2.

| Parameter | Value |
|---|---|
| Discount $\gamma$ | 0.95 |
| GAE $\lambda$ | 0.95 |
| # Timesteps Per Rollout | 64 |
| Train Epochs Per Rollout | 2 |
| # Minibatches Per Epoch | 10 |
| Entropy Coefficient | 0.01 |
| PPO Clip Range | 0.2 |
| Learning Rate | $5 \times 10^{-4}$ |
| Hidden Layer Sizes. | [256,256] |
| # Workers | 8 |
| # Environments Per Worker | 40 |
| Total Timesteps | $8M^{BB}$, $10M^{R}$ |

Table 2: PPO parameters. Superscripts indicate Bouncing Balls (BB) or Reacher (R) environments.

### A.1.3    DATA COLLECTION

We collect *sets* of datasets for use in reward learning and evaluation. Each set contains three sub-datasets for (i) reward model training, (ii) reward model validation, and (iii) reward evaluation. We collect these dataset sets using two policies. The first policy ($\pi_{uni}$) yields a uniform distribution on the action space. The second policy is an expert ($\pi^*$) trained as described in A.1.2. Table 3 gives the sizes of these datasets, which were set such that the validation and train losses were similar during reward learning.

| Environment | Reward Learning Train | Reward Learning Validation | Reward Evaluation |
|---|---|---|---|
| Bouncing Balls | 2,000,000 | 500,000 | 500,000 |
| Reacher | 4,000,000 | 1,000,000 | 1,000,000 |

Table 3: The sizes of the datasets used for reward learning and evaluation for each environment.

### A.1.4 TRANSITION MODEL LEARNING

This section provides details regarding the architecture and training of the learned transition models.

For the Bouncing Balls transition model, we split the state (originally containing the concatenated information for each ball in the scene and the goal position) into states for each ball and predict the motion of each independently. We simplify the task in this manner in order to make learning the transition model easier, but a model that predicts the next state jointly could be learned instead. The absolute velocity of each ball is then passed to a fully-connected network, which outputs a position delta to be applied to the original state. In the case of the controlled ball, the action is also provided to the network. Given this formulation, the network is learning to translate a velocity into a change in position over some time period, optionally applying an acceleration value as well.

For the Reacher environment, we pass the concatenated state and action to a fully-connected network, and again predict changes in the state. We normalize the scale of each state dimension by dividing by its standard deviation across the dataset.

In both cases, we use a mean squared error loss computed between (i) the model output (i.e., state delta) applied to the original state and (ii) the ground truth next state. We train the transition model on data sampled from $\pi_{uni}$, which is common in learning transition models (Ebert et al., 2018). Model and training hyperparameters are given in Table 4.

| Parameter | Value |
|---|---|
| Batch Size | 2048 |
| Hidden Layer Sizes | [256, 256] |
| Activation Function | ReLU |
| Normalization Layer | Batch Norm$^R$, None$^{BB}$ |
| Learning Rate | $(2 \times 10^{-4})^R$, $(5 \times 10^{-4})^{BB}$ |
| Gradient Clip | 1.0 |
| Optimizer | Adam (Kingma & Ba (2014)) |

Table 4: Parameters for learning transition models. For information on datasets used for training see section A.1.3. Superscripts indicate parameters specific to Bouncing Balls environment (BB) or Reacher environment (R).

### A.1.5 REWARD LEARNING

Three methods of learning reward functions were considered in this research: (i) direct regression of per-timestep rewards (REGRESS), (ii) regression on a shuffled dataset matching the evaluation distribution of EPIC (REGRESS OOD), and (iii) preference-based learning of rewards. These methods each fit a fully-connected neural network, and their model and training hyperparameters are provided in table 5. We used early stopping based on validation loss for all models to determine the number of training epochs.

For the Bouncing Balls environment we input into the model a set of manually extracted features including distance of the agent to the goal in the current and next state, the difference in distances between states, an indicator of whether the goal has been reached, squared differences in position of other agents, and others. Features involving squared distances are what make these model susceptible to OOD issues. For the Reacher environment we concatenate the state, action, and next state, and pass that directly into the model. In either case, these features are normalized by subtracting their mean and dividing by their standard deviation.

| Parameter | Value |
|---|---|
| Batch Size | 2048 |
| Hidden Layer Sizes | [256, 256] |
| Activation Function | tanh |
| Learning Rate | $2 \times 10^{-4}$ |
| Gradient Clip | 1.0 |
| Discount $\gamma$ | 0.95 |
| Optimizer | Adam (Kingma & Ba (2014)) |
| Reward Regularization[p] | $0.01^{BB}$, $0.001^{R}$ |
| Trajectory Length[p] | $25^{BB}$, $15^{R}$ |
| # Random Pairs[OOD] | $10^{BB}$, $5^{R}$ |

Table 5: Parameters for reward learning algorithms. For information on datasets used for training see section A.1.3. Superscripts indicate parameters specific to preference learning (p), OOD regression (OOD), Bouncing Balls environment (BB), Reacher environment (R).

### A.1.6 REWARD EVALUATION

**EPIC** We compute the sample-based approximation of EPIC described in section A.1.1 of Gleave et al. (2020). Table 6 gives the parameters we use in computing EPIC. We use the same values of $N_V$ and $N_M$ as Gleave et al. (2020), though use fewer seeds. In our case each seed represents a full execution of the pipeline (data collection, reward model training, reward evaluation), whereas in Gleave et al. (2020) it represents different sub-samplings of a single dataset. We take this approach in order to assess sensitivity to randomness in the data collection and reward learning processes.

| Parameter | Value |
|---|---|
| State Distribution $\mathcal{D}_{\mathcal{S}}$ | Marginalized from $\mathcal{D}$ |
| State Distribution $\mathcal{D}_{\mathcal{A}}$ | Marginalized from $\mathcal{D}$ |
| # Seeds | 5 |
| Samples $N_V$ | 32768 |
| Mean Samples $N_M$ | 32768 |
| Discount $\gamma$ | 0.95 |

Table 6: EPIC parameters.

**DARD** Similarly to EPIC, we compute a sample-based approximation of DARD in the experiments. We sample a batch $B_V$ of $N_V$ $(s, a, s')$ transitions. For computing the expectations $\mathbb{E}[R(s, A, S')]$ and $\mathbb{E}[R(s', A, S'')]$ we sample $N_A$ actions $u$ using a sampling policy (described in Appendix A.1.3). We then sample $N_T$ samples of $S'$ or $S''$, $x'$ and $x''$, conditional on $s$ or $s'$ and $u$. For the last expectation, $\mathbb{E}[R(S', A, S'')]$, we sample actions and next states similarly for $S'$, but repeat the process for $S''$, overall computing the mean over $O((N_A N_T)^2)$ transitions. We compute the dynamics-aware transformation (definition 3) for each transition by averaging over these sampled actions and next states:

$$C_T(R)(s, a, s') = R(s, a, s') + \mathbb{E}[\gamma R(s', A, S'') - R(s, A, S') - \gamma R(S', A, S'')] \qquad (5)$$

$$\approx R(s, a, s') + \frac{\gamma}{N_A N_T} \sum_{i=0}^{N_A} \sum_{j=0}^{N_T} R(s', u_i, x_j'')$$

$$- \frac{1}{N_A N_T} \sum_{i=0}^{N_A} \sum_{j=0}^{N_T} R(s, u_i, x_j') \qquad (6)$$

$$- \frac{\gamma}{N_A^2 N_T^2} \sum_{i=0}^{N_A} \sum_{j=0}^{N_T} \sum_{k=0}^{N_A} \sum_{l=0}^{N_T} R(x_{ij}', u_{kl}, x_{kl}'').$$

Overall this yields an $O(N_V N_A^2 N_T^2)$ algorithm. In our experiments the transition model is assumed to be deterministic, making $N_T = 1$. The actions are sampled from the N-dimensional action space by taking the cross product of linearly-sampled values for each dimension. For example, in the Bouncing Balls environment, we sample 8 actions linearly between $[-5, 5]$ for each of the two dimensions, overall yielding $N_A = 64$. The complexity of computing $\mathbb{E}[R(S', A, S'')]$ could be reduced by instead sampling $N_A$ actions and $N_T$ transitions for each of $x'$ and $x''$.

For the Bouncing Balls environment, we approximate the transition function with a constant velocity model for all agents, implemented in Pytorch and executed on the GPU (Paszke et al., 2019). For the Reacher environment, we use the ground truth simulator, which we execute in parallel across 8 cores. See table 7 for a summary of the DARD parameters used in the experiments.

| Parameter | Value |
|---|---|
| # Seeds | 5 |
| Samples $N_V$ | 100,000 |
| Action Samples $N_A$ | 64[BB], 16[R] |
| Transition Samples $N_T$ | 1 |
| Discount $\gamma$ | 0.95 |

Table 7: DARD parameters. Superscripts indicate parameters specific to the Bouncing Balls (BB) or Reacher (R) environments.

**Policy Evaluation** We compute average episode returns of policies trained on both manually-defined and learned reward models. Training parameters are the same as those used for learning the expert policy (see Table 2). We evaluate policies for 1 million steps in the respective environment, and report the average, undiscounted episode return. Similarly to the other reward evaluation steps, we train one policy for each random seed (of which there are 5).

A.1.7 STANDARD ERROR OF RESULTS

Tables 8 and 9 provide the standard errors in the estimates of the mean distances and episode returns, computed over 5 random seeds.

| Reward Function | 1000 x $D_{DARD}$ | | 1000 x $D_{DARD-L}$ | | 1000 x $D_{EPIC}$ | | 1000 x $D_{\rho}$ | | Episode Return |
|---|---|---|---|---|---|---|---|---|---|
| | $\pi_{\text{uni}}$ | $\pi^*$ | $\pi_{\text{uni}}$ | $\pi^*$ | $\pi_{\text{uni}}$ | $\pi^*$ | $\pi_{\text{uni}}$ | $\pi^*$ | |
| GT | 0.00 | 0.00 | 0.00 | 0.00 | 0.00 | 0.00 | 0.00 | 0.00 | 0.41 |
| SHAPING | 0.00 | 0.00 | 0.00 | 0.00 | 0.00 | 0.00 | 1.56 | 0.31 | 0.20 |
| FEASIBILITY | 0.00 | 0.00 | 0.00 | 0.00 | 3.05 | 0.70 | 1.56 | 0.31 | 0.11 |
| REGRESS OOD | 1.10 | 0.90 | 1.32 | 0.92 | 0.84 | 0.40 | 1.53 | 0.49 | 0.13 |
| REGRESS | 1.70 | 0.92 | 1.38 | 0.94 | 2.07 | 10.1 | 1.56 | 0.37 | 0.13 |
| PREF | 7.20 | 3.20 | 7.52 | 3.15 | 2.10 | 7.74 | 1.57 | 1.14 | 0.13 |

Table 8: Standard error values for Bouncing Balls environment.

| Reward Function | 1000 x $D_{DARD}$ | | 1000 x $D_{DARD-L}$ | | 1000 x $D_{EPIC}$ | | 1000 x $D_{\rho}$ | | Episode Return |
|---|---|---|---|---|---|---|---|---|---|
| | $\pi_{\text{uni}}$ | $\pi^*$ | $\pi_{\text{uni}}$ | $\pi^*$ | $\pi_{\text{uni}}$ | $\pi^*$ | $\pi_{\text{uni}}$ | $\pi^*$ | |
| GT | 0.00 | 0.00 | 0.00 | 0.00 | 0.00 | 0.00 | 0.00 | 0.00 | 1.13 |
| SHAPING | 0.00 | 0.00 | 0.00 | 0.00 | 0.00 | 0.00 | 0.38 | 1.85 | 3.22 |
| FEASIBILITY | 0.00 | 0.00 | 0.00 | 0.00 | 0.68 | 6.65 | 0.36 | 1.61 | 0.68 |
| REGRESS OOD | 1.37 | 15.9 | 1.42 | 6.62 | 1.71 | 4.74 | 0.47 | 2.87 | 0.86 |
| REGRESS | 2.07 | 11.5 | 1.52 | 2.78 | 10.8 | 5.13 | 0.49 | 2.26 | 1.30 |
| PREF | 2.02 | 14.7 | 2.38 | 11.5 | 3.91 | 6.15 | 0.45 | 5.02 | 4.99 |

Table 9: Standard error values for Reacher environment.

| Reward Function | 1000 x $D_{DARD}$ | | 1000 x $D_{EPIC}$ | | 1000 x $D_\rho$ | | Episode Return |
|---|---|---|---|---|---|---|---|
| | $\pi_{\text{uni}}$ | $\pi^*$ | $\pi_{\text{uni}}$ | $\pi^*$ | $\pi_{\text{uni}}$ | $\pi^*$ | |
| GT | 0.00 | 0.00 | 0.00 | 0.00 | 0.00 | 0.00 | $-4.73 \pm 0.48$ |
| REGRESS | 195 | 480 | 14.8 | 34.2 | 6.59 | 32.1 | $-8.02 \pm 0.95$ |

| Reward Function | 1000 x $D_{DARD}$ | | 1000 x $D_{EPIC}$ | | 1000 x $D_\rho$ | | Episode Return |
|---|---|---|---|---|---|---|---|
| | $\pi_{\text{uni}}$ | $\pi^*$ | $\pi_{\text{uni}}$ | $\pi^*$ | $\pi_{\text{uni}}$ | $\pi^*$ | |
| GT | 0.00 | 0.00 | 0.00 | 0.00 | 0.00 | 0.00 | $-2.1 \pm 0.06$ |
| REGRESS | 16.3 | 47.5 | 6.34 | 25.4 | 6.19 | 20.2 | $-2.59 \pm 0.10$ |

Table 10: **Top**: Results for the original Point Maze environment. **Bottom**: Results for the Point Maze environment when using a larger timestep. See the caption of table 1 for how to interpret these results. DARD distances are much closer to those of EPIC in the large-timestep case, indicating that the differences in rewards assigned to nearby transitions has a significant impact on DARD performance. Parameters used for reward learning match those of Gleave et al. (2020). We were unable to exactly reproduce the episode returns of the policies trained on learned reward models. Note that in the larger-timestep case the episode returns will differ due to the change in timestep.

## A.2 ADDITIONAL EXPERIMENTS

### A.2.1 POINT MAZE EXPERIMENTS

In this section we present results for the Point Maze environment considered in Gleave et al. (2020). These results illustrate a disadvantage of DARD. As discussed in section 5, because DARD performs reward comparisons between transitions relative to a transition model, if all the transitions it considers (across all possible actions) have near-identical reward values, it can be sensitive to noise in the reward model. The Point Maze environment fits this description, and DARD performs poorly on it as a result (see table 10). We provide evidence for the hypothesis that the lack of reward-differentiated transitions causes this poor performance by also providing results for an adapted Point Maze environment, which is identical to the original except that we increase the time difference between timesteps by a factor of 5 (such that nearby transitions yield different reward values). In this adapted case, DARD and EPIC distances are significantly closer than in the original environment.

Two additional traits of the environment are relevant. First, the ground truth reward is only a function of the current state and action, but not the next state. As a result, learned reward models tend to ignore the next state, which minimizes the effect of the OOD reward evaluations performed by EPIC. Second, in these experiments (unlike those in section 5) we train reward models to fit the ground truth reward model (as opposed to a shaped version of it). We do this to be consistent with the experiments provided in Gleave et al. (2020); however, this results in the learned reward models exhibiting little shaping (as illustrated by the low distance values of $\mathcal{D}_p$). This is a property of the reward learning algorithm and specific environment in which it is applied, but learned reward models may exhibit significant shaping in general (Fu et al., 2017).

### A.2.2 METRIC SAMPLE SIZE SENSITIVITY

How sensitive are reward comparison metrics to the size of the dataset on which they are computed? The answer depends on the specific environment, reward functions, and comparison metric under consideration. As such, in this section we evaluate the sensitivity to sample size variations of DARD and EPIC in the Bouncing Balls and Reacher environments across a variety of manually defined and learned reward functions.

**Evaluation Procedure** Because we do not have "ground-truth" values that each metric should assume for reward functions in general, we focus instead on the variability of each reward comparison metric for a given sample size. We quantify this variability by computing, for each metric, a 95% confidence interval (CI) of the mean reward distance across different sample sizes. In this context, a smaller CI width indicates that a metric is less sensitive to the random variations in the sampling of the data. We compute the CI by sampling $K = 100$ datasets of size N for varying N from a much

larger population of samples (we use the reward evaluation dataset described in Appendix A.1.3). For each dataset, we compute each metric comparing each reward function to the ground-truth reward function. We compute the CI based on the K resulting distances for each sample size N. For the evaluation datasets, we use $\pi^*$ for the Bouncing Balls environment, and $\pi_{\text{uni}}$ for the Reacher environment. We use $\pi^*$ for Bouncing Balls because the reward is sufficiently sparse that for small sample sizes, N, the metric values are frequently undefined (due to only having samples with zero reward). Using $\pi^*$ instead of $\pi_{\text{uni}}$ increases the fraction of transitions within the goal threshold addressing the problem.

**Reward Functions**  We compare a set of manually defined and learned reward functions. For the manually defined reward functions, we desire a set of rewards that smoothly vary in similarity to the ground-truth reward. To accomplish this, we use a noisy version of the ground-truth reward defined as $R_{\text{noisy}}(s, a, s') = R(s, a, s') + \epsilon$ for $\epsilon \sim \mathcal{N}(0, \sigma)$. By increasing $\sigma$ we produce a sequence of reward functions that are increasingly different from the ground-truth reward. For the learned rewards, we use the REGRESS and REGRESS OOD reward models introduced in Section 5. Recall that REGRESS is trained on transitions sampled from a dataset of demonstrations collected from the environment, whereas REGRESS OOD is trained on the distribution of transitions resulting from sampling actions and next states independently from the starting state. The resulting training distribution matches that used by EPIC in comparing rewards.

**Results**  Fig. 2 shows the results of the experiment, with each row corresponding to a different reward function, and each column corresponding to a different environment (Bouncing Balls on the left and Reacher on the right). We see that the 95% CI width for DARD and EPIC are quite similar for the noisy ground-truth reward functions (top three rows of the figure) across all sample sizes. This indicates that, for these types of reward functions, the two metrics are similarly sensitive to the size of the dataset. Note that as the amount of noise increases so does the width of the CI, but the similarity of results across metrics holds.

For the learned reward models we do observe different behavior between the metrics. In the case of the REGRESS model (4th row from the top), we see that DARD yields significantly smaller CI widths than does EPIC. The reason for this is that EPIC evaluates the reward function at transitions outside its training distribution, which can yield arbitrary values. The additional randomness (with respect to the random sampling of the data subset) incurred in these reward calculations results in more variation in the reward distance, which produces larger CIs. In contrast, DARD avoids these OOD evaluations of the reward, instead staying close to the reward function training distribution where it is more consistent (across data subsets), producing smaller CIs.

For the REGRESS OOD reward model (bottom row), we observe the inverse: EPIC produces smaller CI widths than does DARD. This is because the reward model training distribution matches the one used by EPIC. As a result, the "imagined" transitions of EPIC effectively increase the size of the evaluation dataset producing less variation for a given sample size. DARD also uses "imagined" transitions, but only from the distribution of transitions encountered in practice in the environment.

**Conclusion**  We evaluated the sensitivity of DARD and EPIC to variations in sample size across different environments and reward models. We found that for noisy ground-truth reward functions, the two metrics performed similarly, and that for learned reward functions, the relative sensitivity of the metrics depends on the training distribution of the reward function under consideration.

### A.2.3 RANDOMIZED REWARD FUNCTIONS

The results in Section 5 focus on a special set of reward functions that are relatively similar to the ground-truth reward function. In this section we instead attempt to address the question, how does DARD perform in the full range of reward similarity? We investigate this question by (i) randomly generating a large number of reward functions in the Reacher environment, (ii) optimizing a policy for each reward function, (iii) evaluating that policy against the ground-truth reward function, and (iv) computing the DARD distance between the randomly generated and ground-truth rewards.

**Evaluation Procedure**  The rewards are generated by randomly sampling weights for the three reward components: `distance from goal`, `action magnitude`, `goal-reached`

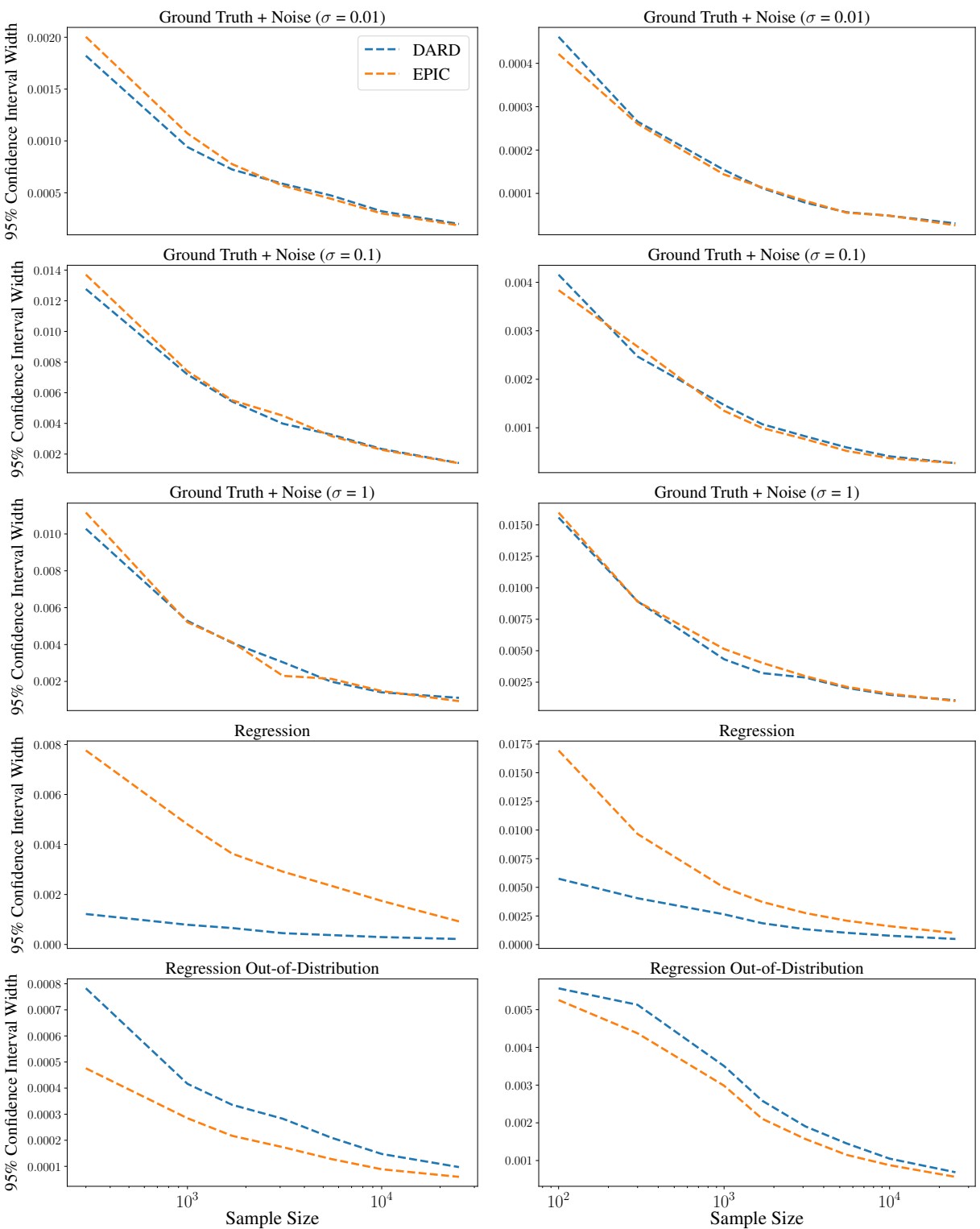

Figure 2: Comparison of the sensitivity of DARD and EPIC to variations in the size of the dataset.
**Left**: Results for the Bouncing Balls environment. **Right**: Results for the Reacher environment. See Appendix A.2.2 for a detailed description of the figure and analysis of the results.

indicator. Weights for the first two components are sampled uniformly between $[0, 1]$. The reason we disallow negative weights for these components is that they make policy optimization challenging or ill-posed: in the `distance from goal` case it creates a challenging exploration problem, and in the `action magnitude` case it encourages the policy to output increasingly large actions. Weights for the `goal-reached indicator` component are sampled uniformly from $[-1, 1]$. The reward function is then computed as a linear combination of the reward components weighted by these sampled values. We sample 128 reward functions in total. Policy optimization, reward evaluation, and policy evaluation are performed as described in Appendix A.1.

**Results**  Fig. 3 visualizes the results from the experiment, computing DARD on the random ($\pi_{\mathrm{uni}}$) and expert ($\pi^*$) coverage distributions. We see that in both cases DARD is predictive of policy return as indicated by the association between small DARD distances and large policy return. The DARD distance and policy return are both heavily dependent on the sign of the `goal-reached indicator` weight, which we visualize by coloring points based on the sign of this weight. When the sign is negative, the policy learns to avoid getting too close to the goal, thereby dramatically reducing its expected return under the ground-truth reward. The $\pi_{\mathrm{uni}}$ plot shows policy return decaying more quickly as a function of DARD distance than does the plot of $\pi^*$. The reason for this is that in the $\pi_{\mathrm{uni}}$ case, the action magnitude contributes much more significantly to the DARD distance than in the $\pi^*$ case (because $\pi^*$ generally consists of small actions near the goal state).

There are some outliers in the plot, which seem to result from extreme combinations of weight values that can cause policy optimization issues. For example, cases of low DARD distance and *low* policy return in general tend to have small `distance from goal` weights and large, positive `goal-reached indicator` weights. In these cases, the reward is quite sparse, causing problems for policy optimization (based on prior experiments, the policy does eventually achieve high expected return, but only with more training timesteps). In the $\pi_{\mathrm{uni}}$ case, instances of large DARD distance and *high* policy return tend to result from extremely small `action magnitude` weights and large, positive `goal-reached indicator` weights. In these cases, the optimized policy takes large magnitude actions to quickly reach the goal state. This incurs a large penalty over the first few timesteps of the episode, after which the agent remains mostly stationary, accruing the goal-reached reward. This trend does not hold for $\pi^*$, which demonstrates DARD's sensitivity to the coverage distribution.

**Conclusion**  In this section, we evaluated DARD on a large number of random reward functions in the Reacher environment, many of which differ significantly from the ground-truth reward. Results demonstrate that DARD maintains its predictiveness of policy return in this setting across both coverage distributions.

## A.3  ADDITIONAL DISCUSSION

### A.3.1  THE IMPACT OF TRANSITION MODEL ERRORS ON DARD

In this section, we discuss how errors made by the transition model can impact the performance of DARD. We organize our discussion of this topic around different cases of reward equivalence and different types of errors that a transition model might make. The high-level summary of the following discussion is that errors in the transition model only impact the performance of DARD on reward functions that are not equivalent everywhere (case 2 below), and in this case DARD may produce both false negative and false positive equivalences as a result of transition model errors.

**Case 1: Equivalent Reward Functions**  Consider two reward functions that are equivalent *across the full space of transitions* accounting for reward shaping (this includes the case of identical rewards). The proof of Proposition 2 does not rely on any proprieties of the transition model. As a result, we can conclude that even arbitrary errors in the transition model will not result in two everywhere-equivalent reward functions being considered different. This logic indicates that errors in the transition model may only impact the performance of DARD when the reward functions are not everywhere-equivalent as discussed next.

**Case 2: Different Reward Functions**

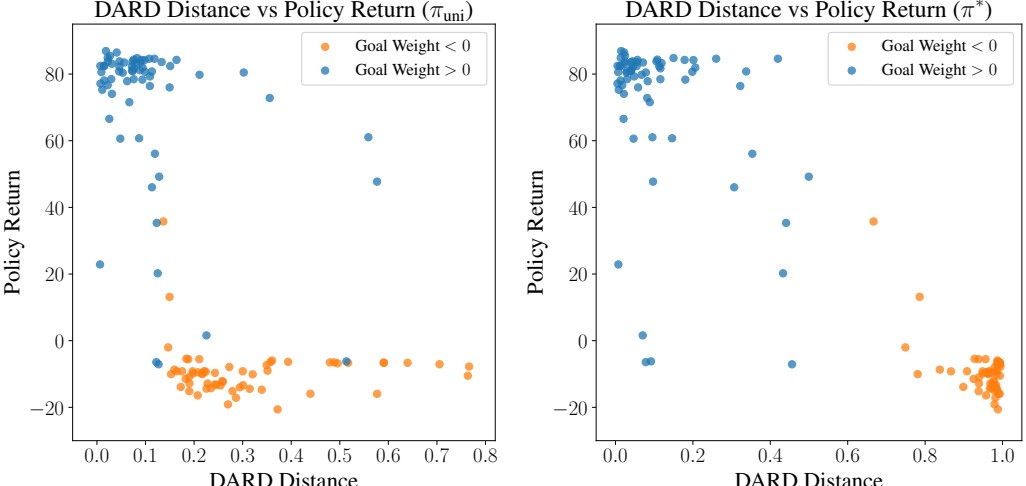

Figure 3: Comparison of DARD distance with policy return for randomized reward functions in the Reacher environment. Each point in these scatter plots corresponds to a different randomly generated reward function. We train a policy on each of these reward functions, and then evaluate that policy against the ground-truth reward function. The resulting mean policy return is plotted on the y-axis. On the x-axis we plot the DARD distance computed with respect to the $\pi_{\text{uni}}$ (**Left**) and $\pi^*$ (**Right**) coverage distributions. See Appendix A.2.3 for an analysis of the results.

SUBCASE A: REWARDS ARE EQUIVALENT ON ALL *Realizable* TRANSITIONS    Given the ground truth transition model, DARD would correctly identify the two rewards as equivalent (with respect to realizable transitions). However, if the transition model erroneously samples unrealizable transitions on which the rewards differ, the transformation term of Eq. (3) could assume arbitrary values, thereby resulting in the two rewards being considered different. This might occur, for example, when dealing with learned reward functions for which sampled next states are OOD with respect to their training distribution.

SUBCASE B: REWARDS DIFFER ON SOME REALIZABLE TRANSITIONS    Given the ground truth transition model and sampling of the full action space, DARD would correctly identify the two rewards as different (assuming states producing transitions on which the rewards differ exist in the dataset). However, if the transition model is incorrect in such a way that it never samples those transitions on which the reward models differ, DARD *may* produce false positive equivalences.

For example, suppose a transition model is learned for an autonomous driving environment. Because collisions are rare in this environment, the transition model might never sample transitions involving collisions. As a result, two reward models that differ only with respect to transitions involving collisions *may* be considered equivalent by DARD. In the preceding text, we say that DARD *may* produce false positive equivalences in this case because even with an erroneous transition model, if the dataset on which the rewards are compared itself contains transitions on which the rewards differ DARD will still recognize them as different (due to the $R(s, a, s')$ term of Eq. (3)). Continuing the autonomous driving example: if the dataset itself contains collisions, then the rewards will be identified as different even if the transition models do not sample collisions. This case illustrates the importance of using a comprehensive coverage distribution in comparing reward functions.

### A.3.2    ANALYSIS OF DARD-L IN REACHER WITH $\pi^*$ COVERAGE DISTRIBUTION

This section provides a detailed analysis of the results for DARD-L in the Reacher environment when using the coverage distribution of the expert policy.

Unlike the $\pi_{\text{uni}}$ coverage distribution, the states sampled from the expert distribution in the Reacher environment generally have small distances between the end-effector and goal location. As a result, the reward is dominated by whether or not the end-effector is within the threshold distance at which

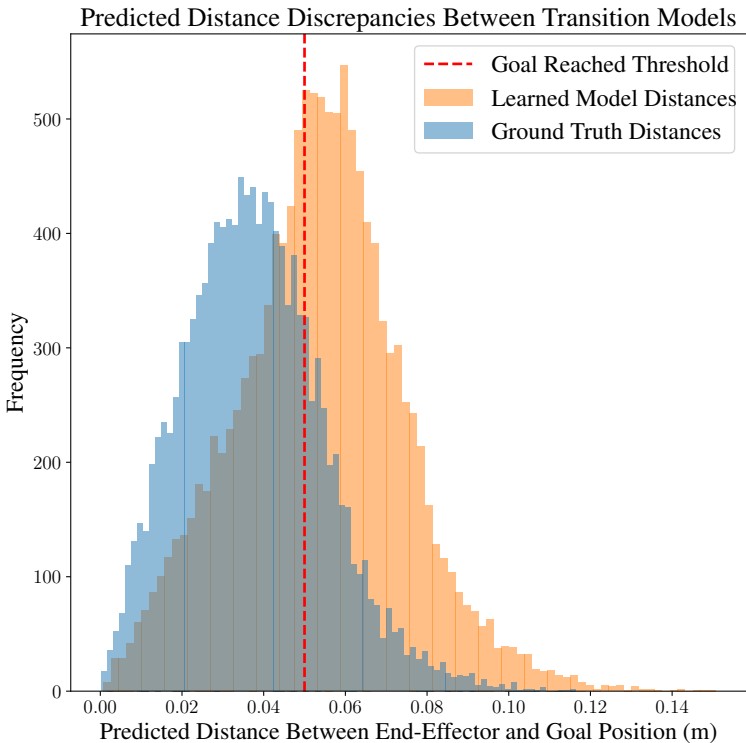

Figure 4: This plot shows histograms of the ground truth and predicted next state displacements between the end-effector and goal location in the Reacher environment. These displacements are computed on 200 transitions for which the learned transition model yields transformed reward values that most differ from those of the ground truth transition model. The displacements predicted by the learned transition model are consistently larger than those of the ground truth model, and are often beyond the threshold at which the "goal reached" reward is assigned.

the "goal reached" reward is given. This causes the reward functions to be sensitive to small errors in the predicted state because those errors can move the end-effector in or out of range of the goal.

On average, the learned transition model predicts larger next state end-effector displacements than the ground truth transition model. This can be seen in Fig. 4 where we plot the histogram of ground truth and predicted displacements over a set of transitions for which the transformed reward differs the most between the ground truth and learned transition models. These erroneous, larger predicted distances result from using a mean squared error loss, which causes displacements near zero to contribute a minuscule amount to the loss.

As a result, the next states predicted by the learned transition model are on average easier to assign rewards to than those of the ground-truth transition model: they are more clearly beyond the limit of the "goal reached" threshold, and (due to being farther away) reward shaping (which is easier to predict) constitutes a larger fraction of the reward. Overall this results in the learned transition model yielding lower DARD distances than those of the ground truth transition model. This is an instance of Case 2.B in Appendix A.3.1: states at which the reward models differ more are sampled relatively less frequently due to errors in the learned transition model.

### A.3.3 APPLICATIONS OF REWARD COMPARISON METRICS

This section discusses two potential applications of reward comparison metrics.

**Benchmarking Reward Learning Algorithms**  Suppose we have a large amount of (potentially suboptimal) demonstration data from which we would like to learn a reward function using one of a variety of potential IRL algorithms. Additionally, suppose that we have (through an expen-

sive process) manually labeled rewards in a relatively small set of demonstrations. Furthermore, assume that evaluating a policy online involves an expensive or dangerous process, and that learning a policy for this environment is itself challenging (e.g., sometimes fails or produces policies of highly-variable quality). Note that this setting might occur in practice in physical robotic domains such as autonomous driving or robotic manipulation.

In summary, we would like to know which of the set of possible IRL algorithms produces the best reward function using the large set of demonstrations, but we would prefer to avoid policy learning and evaluation. One way to accomplish this goal would be to learn a "benchmark" reward model on the small set of manually labeled rewards (e.g., through supervised learning as in Cabi et al. (2019)), and then to compare that with the rewards produced by the various IRL algorithms. If a reward function learned via IRL is similar to the benchmark reward function, this indicates that it is similar to the reward function manually defined by experts, and should therefore be preferred. In this way, we can evaluate the different IRL methods we are considering, and choose the best for our problem. Furthermore, this approach could be applied in creating an open benchmark enabling the comparison of reward learning methods.

**Validating Learned Reward Functions** Suppose that we have deployed a system that uses a learned reward function. With deployed learning systems, it is often necessary to continuously collect data and retrain models in order to account for changing data distributions arising from a variety of sources (Breck et al., 2017). As such, we would like to frequently retrain the reward function on newly-collected data; however, we would also like to ensure that the reward function has not changed "too much" from previous versions that we know to be of high quality (through expensive deployment). Similar to the previous application, suppose that online policy evaluation is expensive or dangerous, and that policy optimization is challenging.

Given this problem setting, we could use reward comparisons as follows. First, learn a new reward function on all the data collected so far. Second, compare that reward function with the old reward function on the validation dataset of the old reward. If the comparison yields a small distance between rewards, this indicates that the reward functions are likely quite similar, thereby providing support for continuing the process of deploying the new reward function. Alternatively, if the comparison yields a large distance between rewards, this indicates that the new reward is potentially quite different from the old one, and that we should carefully inspect the cases where the rewards differ before proceeding with deployment. In this way, we have defined (part of) a validation procedure for continuously learning and deploying reward functions.

## A.4 Proofs

### A.4.1 Comparing Learned Reward Functions

**Proposition 3** (The dynamics-aware transformed reward is invariant to shaping). *Let $R : \mathcal{S} \times \mathcal{A} \times \mathcal{S} \to \mathbb{R}$ be a reward function, and $\Phi : \mathcal{S} \to \mathbb{R}$ a potential function. Let $\gamma \in [0, 1]$ be a discount rate, and $\mathcal{D}_{\mathcal{S}} \in \Delta(\mathcal{S})$ and $\mathcal{D}_{\mathcal{A}} \in \Delta(\mathcal{A})$ be distributions over states and actions, and $T : \mathcal{S} \times \mathcal{A} \to \Delta(\mathcal{S})$ be a conditional distribution over next states. Let $R'$ denote $R$ shaped by $\Phi : R'(s, a, s') = R(s, a, s') + \gamma \Phi(s') - \Phi(s)$. Then the dynamics-aware transformations of $R'$ and $R$ are equal: $C_T(R') = C_T(R)$.*

*Proof.* Let $s, a, s' \in \mathcal{S} \times \mathcal{A} \times \mathcal{S}$. Then by substituting in the definition of $R'$ and using linearity of expectation:

$$C_T(R')(s, a, s') \triangleq R'(s, a, s') + \mathbb{E}[\gamma R'(s', A, S'') - R'(s, A, S') - \gamma R'(S', A, S'')] \quad (7)$$

$$= (R(s, a, s') + \gamma \Phi(s') - \Phi) \quad (8)$$
$$+ \mathbb{E}[\gamma R(s', A, S'') + \gamma^2 \Phi(S'') - \gamma \Phi(s')]$$
$$- \mathbb{E}[R(s, A, S') + \gamma \Phi(S') - \Phi(s)]$$
$$- \mathbb{E}[\gamma R(S', A, S'') + \gamma^2 \Phi(S'') - \gamma \Phi(S')]$$
$$= R(s, a, s') + \mathbb{E}[\gamma R(s', A, S'') - R(s, A, S') - \gamma R'(S', A, S'')] \quad (9)$$
$$+ (\gamma \Phi(s') - \Phi(s)) - \mathbb{E}[\gamma \Phi(s') - \Phi(s)]$$
$$+ \mathbb{E}[\gamma^2 \Phi(S'') - \gamma \Phi(S')] - \mathbb{E}[\gamma^2 \Phi(S'') - \gamma \Phi(S')]$$
$$= R(s, a, s') + \mathbb{E}[\gamma R(s', A, S'') - R(s, A, S') - \gamma R'(S', A, S'')] \quad (10)$$
$$= C_T(R)(s, a, s'). \quad (11)$$

$\square$

**Lemma A.1** (The Pearson distance $\mathcal{D}_\rho$ is a pseudometric). *(Lemma 4.5 from Gleave et al. (2020)) Namely, for $\mathcal{X}$ a set of random variables, $\mathcal{D}_\rho$ satisfies the following properties:*
 ***Identity:*** $\mathcal{D}_\rho(X, X) = 0 \ \forall X \in \mathcal{X}$,
 ***Symmetry:*** $\mathcal{D}_\rho(X, Y) \mathcal{D}_\rho(Y, X) \ \forall X, Y \in \mathcal{X}$, and
 ***Triangle Inequality:*** $\mathcal{D}_\rho(X, Z) \leq \mathcal{D}_\rho(X, Y) + \mathcal{D}_\rho(Y, Z) \ \forall X, Y, Z \in \mathcal{X}$.

*Proof: See Gleave et al. (2020)*

**Proposition 4** (The DARD distance metric is a pseudometric). *Let $S$, $A$, $S'$ be random variables jointly following some coverage transition distribution, and let $T$ be the transition model of the corresponding environment. We define the DARD pseudometric between rewards $R_A$ and $R_B$ as follows:*

$$D_{DARD}(R_A, R_B) = D_\rho\Big(C_T(R_A)(S, A, S'), C_T(R_B)(S, A, S')\Big) \quad (12)$$

*Proof.* This proof closely follows the proof of Theorem 4.7 from Gleave et al. (2020). Namely, the result follows from direct application of Lemma A.1.

Let $R_A$, $R_B$, and $R_C$ be reward functions mapping from transitions $\mathcal{S} \times \mathcal{A} \times \mathcal{S}$ to real numbers $\mathbb{R}$.

**Identity:** It immediately follows that
$D_{\text{DARD}}(R_A, R_A) = D_\rho\Big(C_T(R_A)(S, A, S'), C_T(R_A)(S, A, S')\Big) = 0$ since $D_\rho$ is a pseudometric, $D_\rho(X, X) = 0 \ \forall X \in \mathcal{X}$.

**Symmetry:** This again immediately follows from the fact that $D_\rho$ is a pseudometric. We have that:

$$D_{\text{DARD}}(R_A, R_B) = D_\rho\Big(C_T(R_A)(S, A, S'), C_T(R_B)(S, A, S')\Big) \quad (13)$$

$$= D_\rho\Big(C_T(R_B)(S, A, S'), C_T(R_A)(S, A, S')\Big) \quad (14)$$

$$= D_{\text{DARD}}(R_B, R_A) \quad (15)$$

since $D_\rho(X, Y) = D_\rho(Y, X)$ because $D_\rho$ is a pseudometric.

**Triangle Inequality:** This again immediately follows from the fact that $D_\rho$ is a pseudometric. We have that:

$$D_{\text{DARD}}(R_A, R_C) = D_\rho\Big(C_T(R_A)(S, A, S'), C_T(R_C)(S, A, S')\Big) \quad (16)$$

$$\leq D_\rho\Big(C_T(R_A)(S, A, S'), C_T(R_B)(S, A, S')\Big) \quad (17)$$

$$+ D_\rho\Big(C_T(R_B)(S, A, S'), C_T(R_C)(S, A, S')\Big) \quad (18)$$

$$= D_{\text{DARD}}(R_A, R_B) + D_{\text{DARD}}(R_B, R_C) \quad (19)$$

since $D_\rho(X, Z) \leq D_\rho(X, Y) + D_\rho(Y, Z)$ because $D_\rho$ is a pseudometric. $\square$

