# OpenReview forum: "Dynamics-Aware Comparison of Learned Reward Functions"
_ICLR.cc/2022/Conference — ICLR 2022 Spotlight_

### Official Review · Reviewer_6Mwn · 2021-11-02

**Correctness:** 4
**Technical Novelty And Significance:** 3
**Empirical Novelty And Significance:** 4
**Recommendation:** 8
**Confidence:** 3

**Main Review:**

This paper makes progress on the important problem of comparing reward functions by addressing a significant shortcoming of the EPIC distance. The paper is clearly written and well-motivated. The theoretical results seem sound based on a quick skim of the proofs. The method is evaluated empirically on a wide range of reward models, and claims H1-H3 seem well-supported by the results.

The advantage of DARD over EPIC comes at the cost of assuming knowledge of the transition dynamics, which is a limitation for scaling the method to more complex settings where an environment simulator may not be available. It would be great to see how well DARD works for learned transition models as suggested in the conclusion, though I understand this is out of scope for this paper.

Here are some suggestions for improving the paper:
* The fact that DARD does not perform well on environments like PointMaze where nearby transitions always have similar reward values is an important limitation, and I would suggest adding the PointMaze experiment to the main text of the paper.

* Define a feasibility reward model for the Reacher environment as well, to demonstrate that this shortcoming of EPIC is not specific to one environment.

* To further support hypothesis H3 that DARD is effective across diverse coverage distributions, it may be useful to also test it on a coverage distribution induced by a non-random suboptimal policy.

* It would be good to see the two methods compared on more than two environments, e.g. other MuJoCo environments.

* Clarify the difference between $D_\rho$ and $D_p$, which seem to both be defined as the Pearson distance



**Summary Of The Paper:**

This paper proposes a novel method for comparing reward functions without policy optimization, the Dynamics-Aware Reward Distance. DARD improves on the state-of-the-art EPIC distance by using an approximate transition model to evaluate reward functions on transitions close to the training distribution, while EPIC evaluates on arbitrary transitions (which can be infeasible). The authors prove that DARD is also invariant to reward shaping, and experimentally demonstrate on the Bouncing Balls and Reacher environments that DARD is a better predictor of policy return than EPIC and works for coverage distributions induced by either random or expert actions.

**Summary Of The Review:**

This paper makes progress on the important problem of comparing reward functions by introducing a new reward distance that addresses a significant shortcoming of the EPIC distance. The results are significant and novel, and I would recommend accepting the paper.

---

> ### Author Response · Authors · 2021-11-16
> **Response to Reviewer 6Mwn**
>
> Thank you for reviewing the paper! We appreciate your positive comments regarding the motivation for the research and the algorithm itself, as well as your helpful suggestions for how we can improve the paper! We have responded to your recommendations below by quoting the relevant section of your review.
>
> > It would be great to see how well DARD works for learned transition models
> - We agree that the ability to apply DARD in conjunction with learned transition models would significantly increase its applicability, and towards understanding the performance of the algorithm in that setting, have added results to the paper applying DARD with learned transition models in the environments already considered. Please see section 5 for a discussion of the results, and section A.1.4 for details of transition model training.
> - We believe additional research that considers the use of DARD with learned transition models in high-dimensional (e.g., visual) settings is of course still necessary, but we hope the additional experiments provide some initial insight.
> - As we mentioned in our response to reviewer vDTh, the use of a *learned* transition model with DARD is an important special case of applying DARD with an *erroneous* transition model. We have added section A.3.1 in order to categorize and explain the types of errors DARD might make as a result of errors in the transition model.
>
> > The fact that DARD does not perform well on environments like PointMaze where nearby transitions always have similar reward values is an important limitation, and I would suggest adding the PointMaze experiment to the main text of the paper.
> - We agree that this is an important result that the reader should be made aware of. Given the additional text regarding the use of learned transition models, we believe it would be challenging to fit the PointMaze results in the main body of the paper. We additionally believe these results should be accompanied by the supporting discussion currently in the appendix if possible. For these reasons we would prefer to keep these results in section A.2.1, but are open to trying to edit and reformat the paper to include them in the main body if you feel strongly about this change.
>
> > Define a feasibility reward model for the Reacher environment as well, to demonstrate that this shortcoming of EPIC is not specific to one environment.
> - This is an excellent suggestion, thank you! We are working on implementing this, and hope to provide results before the end of the review period.
>
> > To further support hypothesis H3 that DARD is effective across diverse coverage distributions, it may be useful to also test it on a coverage distribution induced by a non-random suboptimal policy.
> - We agree that evaluating across a diverse set of coverage distributions is important, and are working on implementing this.
>
> > It would be good to see the two methods compared on more than two environments, e.g. other MuJoCo environments.
> - While we believe that the results on the two environments considered do provide substantial evidence regarding DARD’s ability to compare learned reward functions, we agree that results on additional environments would be beneficial. If we have sufficient time we will run experiments on an additional environment, though we do plan on prioritizing some of the other experiments you and the other reviewers suggested.
>
> > Clarify the difference between Dρ and Dp’, which seem to both be defined as the Pearson distance
> - Thank you for identifying this issue. We intended to use Dρ throughout, and have changed the paper to reflect this.

---

> > ### Author Response · Authors · 2021-11-19
> > **Response to Reviewer 6Mwn**
> >
> > We have made changes since our previous response addressing your comment regarding the feasibility model for the Reacher environment:
> > > Define a feasibility reward model for the Reacher environment as well, to demonstrate that this shortcoming of EPIC is not specific to one environment.
> > - We have updated the paper to include results for a feasibility reward model for the Reacher environment. The distances assigned by DARD and EPIC between this model and the ground-truth model are as expected, with DARD assigning 0 distance and EPIC assigning large distances. Please see the updated Table 1 for details, and thanks again for the suggestion.

---

> > ### Comment · Reviewer_6Mwn · 2021-11-24
> > **Thank you for the response**
> >
> > Thank you for addressing my suggestions for the paper.
> > * It's great to see the additional experiments with learned transition models and the feasibility model for Reacher, which provide further support for the claims of the paper.
> > * I appreciate that the changes are highlighted in the revised version of the paper, which makes it easy to follow the updates.
> > * I understand that it's hard to fit the Point Maze results in the main text. I think it would be good to highlight the paragraph in the paper that summarizes the Point Maze results, e.g. by labeling the paragraph as "Limitations".

---

> > > ### Author Response · Authors · 2021-11-29
> > > **Response to Reviewer 6Mwn**
> > >
> > > Thank you for following up and for the additional suggestion! We will add a "Limitations" heading to the paragraph describing the Point Maze results if the paper is accepted (because we can not submit further revisions at this point in the review process).

---

### Official Review · Reviewer_4Nse · 2021-11-02

**Correctness:** 3
**Technical Novelty And Significance:** 2
**Empirical Novelty And Significance:** 3
**Recommendation:** 5
**Confidence:** 3

**Main Review:**

Strengths:
1. DARD does not relies on the input datasets, and therefore avoids unrelied reward values due to inconsistent samples.
2. The numerical results show that DARD is more powerful compared to EPIC. In particular, the experiments demonstrate that samples "out of distribution" would harm the performance of EPIC significantly.


Weaknesses:
1.  Compared to EPIC, DARD needs access to the transition model $T$ when computing the reward function, which may fail in many practical cases. Even assuming the transition model $T$ is accessable, DARD requires additional samples, which may worse the sample efficiency.
2.  The theorectical intuition of DARD is not clear. In intuition, the samples "out of distribution" leads to inaccuracy. Then the questions are that: how to define a sample "out of distribution" and how it harms the performance of EPIC? There are many measurements to describe the inconsistency between the offline distribution and online distribution and I think it is possible to derive some theorectical guarantee using these arguments.


**Summary Of The Paper:**

The paper proposes a new reward pseudometric (DARD), which is invariant to reward shaping and computationally efficient. Assuming access to the transition model of the environment, DARD could avoid the error due to samples out of the training distribution. The numerical results in simulated physical domains show that DARD outperforms previous methods.

**Summary Of The Review:**

Overall, the paper is well written. My current score is 5 and I look forward to further discussion with the authors.

---

> ### Author Response · Authors · 2021-11-16
> **Response to Reviewer 4Nse**
>
> Thank you for reviewing the paper and providing feedback! We have responded to your comments below by quoting the relevant section of your review.
>
> > The theoretical intuition of DARD is not clear. In intuition, the samples "out of distribution" leads to inaccuracy. Then the questions are that: how to define a sample "out of distribution" and how it harms the performance of EPIC?
> - We would like to try to clarify what we mean when we say a sample is “out of distribution”. When we say “out of distribution”, we are referring to (S,A,S’) tuples that are dynamically impossible and would never be observed: i.e., have probability zero under a transition function p(S’|S,A). We agree that this terminology may have been confusing, and have changed the paper to use "unrealizable" in cases where we believe it clarifies our intent. The remainder of our reply to this comment explains what we mean in more detail:
> - In our setting, we are comparing reward functions that take as input the current state (S), action (A), and next state (S’). We are interested in reward functions that have been learned from data sampled from the *joint* distribution over (S, A, S’) transitions.
> - EPIC is designed to compute reward values on (S, A, S’) triples that may not occur under the joint distribution used during reward training (i.e., impossible transitions). Specifically, EPIC samples the state S independently of action A and next state S’, which disregards their strong interdependence, leading to a significantly different distribution than that observed during reward function training.
> - That is the sense in which EPIC computes rewards “out of distribution”: it computes rewards on (S, A, S’) triples that might have zero or very low probability under their joint distribution and as such do not occur in the reward function training distribution. As a result, the reward values can in general be highly unreliable, and we show in the experiments (section 5) that this can render EPIC a poor predictor of policy return and therefore ineffective at comparing learned reward functions.
>
> > Compared to EPIC, DARD needs access to the transition model T when computing the reward function, which may fail in many practical cases.
> - It is true that DARD requires access to a transition model; however, this is part of the motivation for DARD: if we do have access to additional information, like a transition model, can we exploit this information to develop a better metric than EPIC? While this certainly does not apply to all problem settings as you mention, we propose our DARD method for the situations where either (1) the transition function is known, or (2) a transition model could be learned. In these cases we propose using DARD over EPIC.
> - While reasonable (potentially approximate) transition models are available in many domains (particularly physical ones), in certain domains (e.g., high-dimensional ones involving vision observations) this is not the case. One method of addressing the requirement of access to a transition model in these high-dimensional domains is to instead learn the transition model.
> - In response to your comment and those of other reviewers, we have added results to the paper for DARD when using learned transition models (see section 5 and appendix A.1.4). While these results are for relatively low-dimensional environments, we hope they provide additional insight into the performance of DARD when using learned transition models, and illustrate how DARD might be used when an (approximate) transition model is not immediately available for an environment. Additionally, we added a section discussing the impact of errors in the transition model (for example, those that a learned transition model might make) on the performance of DARD (please see appendix A.3.1).
>
> > Even assuming the transition model T is accessable, DARD requires additional samples, which may worse the sample efficiency.
> - Thank you for raising this point. We are working on an experiment comparing the sample efficiency of the different metrics, which we hope to finish prior to the review deadline.

---

> > ### Author Response · Authors · 2021-11-17
> > **Response to Reviewer 4Nse**
> >
> > We have made changes since our previous response addressing your comment regarding the sample efficiency of DARD:
> >
> > > Even assuming the transition model T is accessable, DARD requires additional samples, which may worse the sample efficiency.
> > - We have added an experiment that investigates the sensitivity of DARD and EPIC to the amount of data used in comparing reward functions, which we describe below, but we would first like to clarify that DARD and EPIC have identical data requirements: both operate on a fixed dataset of transitions, and the number of samples used does not *necessarily* differ.
> > - In response to your comment, we added an experiment that investigates how sensitive DARD and EPIC are to the amount of data used in comparing rewards. The conclusion is that for the manually-defined reward functions considered, DARD and EPIC exhibit similar sensitivity, however, for learned reward functions the sensitivity of the metrics depends on the training distribution of the reward functions. When training rewards from demonstration data, DARD exhibits less sensitivity to the random sampling of the data. Please see Appendix A.2.2 for visualizations of the results and a detailed discussion.
> > - We believe these results indicate that DARD does not have worse sample efficiency than EPIC (where we interpret “sample efficiency” to mean “sensitivity of the metric to the amount data” because for any sample size the expected metric value is the same) for many reward functions of practical interest.

---

### Official Review · Reviewer_LfGd · 2021-11-02

**Correctness:** 4
**Technical Novelty And Significance:** 3
**Empirical Novelty And Significance:** 3
**Recommendation:** 8
**Confidence:** 3

**Main Review:**

* Pros
  * The paper clearly identifies an issue with the prior work (EPIC) and proposes an appropriate solution.
  * The paper is very well-written.
  * The experiments are well-designed.
* Cons
  * The empirical results could be more comprehensive.
  * The proposed method is a bit incremental to EPIC.

---

* The paper motivates the problem well by clearly identifying the issue with the prior work (EPIC) and proposes a reasonable solution that takes into account the actual dynamics of the environment.
* The experiments (various manually designed / learned reward functions) are carefully designed to verify the hypotheses of the paper. I also appreciate the authors for acknowledging the limitation of the proposed method with smooth reward functions.
* The presentation of the paper is excellent. The problem is introduced nicely with appropriate preliminaries, and the main idea is illustrated with a good motivating example (Figure 1).
* Although the empirical results are good, it could be more comprehensive if the paper showed how the amount of the data affects the results, because the proposed method seems a bit more sensitive to the the amount of transitions in the dataset, because this is how the proposed method taking into account the dynamics. For example, it could be interesting to see how the results change as the amount of data available is varied.
* Although the idea of considering dynamics for reward comparison in EPIC is novel, the overall method seems a little incremental because the method heavily builds upon EPIC in that it uses a similar canonicalization step followed by the same Pearson distance measure.

**Summary Of The Paper:**

This paper considers a reward comparison method in RL, where the goal is to evaluate how close an alternative reward function is to the ground truth reward function in terms of how well the set of optimal policies for each reward function is aligned. The proposed method (DARD) builds upon EPIC [Gleave et al.] and addresses an issue of EPIC by taking into account the actual dynamics of the environment when canonicalizing/transforming reward function as opposed to considering all possible (s, a, r') triples in EPIC. This allows DARD to be more robust to out-of-distribution transitions compared to EPIC while still being invariant to reward shaping like EPIC. The empirical results on Bouncing Balls and Reacher environments show that DARD can represent the reward distance more reliably EPIC does.

**Summary Of The Review:**

Although the overall method is still quite similar to EPIC, this paper proposes a reasonable solution to overcome a clear limitation of EPIC. In addition, the experiments are quite well-designed, and the paper is very well-written.


---
Update after the rebuttal: Thank you for adding experiments with learned transition models. The results look good. Since my major concern is addressed, I increased my score.

---

> ### Author Response · Authors · 2021-11-16
> **Response to Reviewer LfGd**
>
> Thank you for reviewing the paper; we appreciate your positive feedback and suggestions for improvement! We have responded to your recommendations below by quoting the relevant section of your review.
>
> > Although the empirical results are good, it could be more comprehensive if the paper showed how the amount of the data affects the results, because the proposed method seems a bit more sensitive to the the amount of transitions in the dataset, because this is how the proposed method taking into account the dynamics.
> - Thank you for the suggestion! We are working on an experiment that evaluates the performance of DARD and the baseline methods as the amount of data is varied.
> - In the vein of adding more empirical results, we implemented DARD with a learned transition model in the environments considered in the main body of the paper, and provide results and analysis in section 5.
>
> Please see our general response for a discussion of our view of the contribution of the research relative to prior work.

---

> > ### Author Response · Authors · 2021-11-17
> > **Response to Reviewer LfGd**
> >
> > We have made changes since our previous response addressing your feedback regarding the need for additional empirical results:
> > > Although the empirical results are good, it could be more comprehensive if the paper showed how the amount of the data affects the results, because the proposed method seems a bit more sensitive to the the amount of transitions in the dataset, because this is how the proposed method taking into account the dynamics.
> > - We have added an experiment investigating how sensitive DARD and EPIC are to the amount of data used in comparing rewards. The conclusion is that for the manually-defined reward functions considered, DARD and EPIC exhibit similar sensitivity, however, for learned reward functions the sensitivity of the metrics depends on the training distribution of the reward functions. When training rewards from demonstration data, DARD exhibits less sensitivity to the random sampling of the data. Please see Appendix A.2.2 for visualizations of the results and a detailed discussion. We hope this experiment helps to improve understanding the performance of DARD in comparing reward functions, and addresses your feedback regarding the need for additional empirical results.
> > - We would also like to clarify: DARD takes into account the dynamics of the environment through the use of a transition model. When this transition model is known, DARD does not rely on the transitions in the dataset for sampling actions and next states.

---

### Official Review · Reviewer_vDTh · 2021-11-05

**Correctness:** 3
**Technical Novelty And Significance:** 3
**Empirical Novelty And Significance:** 3
**Recommendation:** 6
**Confidence:** 3

**Main Review:**

The idea of comparing reward function without training policies is very appealing for Reinforcement Learning. The authors take a previous method (EPIC) and improve it by making it consider only feasible state transitions. As a results the distance between reward function become more aligned with the trained policies scores.

The proposed idea is techincally sound and well presented. It is an incremental change based on a previous method, but the authors explain in which possible scenarios it would provide benefits. The evaluation is scritly oriented to validate this claim. The authors compare few reward functions crafted for these scenarios. In my opinion, it is hard to draw (or support) the conclusion from 5 reward functions. It would be very beneficial if the authors could provide a scenario where the DARD compares hundreds (or thousands) of arbitrarily chosen reward functions, and based on its results the authors could train a policy that outperforms the other. This is actually related to a high level comment that I have. Reading the paper, it was not easy for me to understand how the proposed method can be used in a large scale reward designing process. I think that the theme of "reward designing based on DARD for a very complex (or unknown) problem" could make the paper more impactful.

The method assumes access to the dynamics model, and the authors state that this can be approximated (or learned) as well. I believe that an example based on learned dynamics model would make the paper much better.


**Summary Of The Paper:**

The paper proposes a new reward pseudometric called Dynamics Aware Reward Distance (DARD) which uses approximate transition model of the environment to compare reward functions while being indifferent to reward shaping. Previous work Equivalent Policy Invariant Comparison (EPIC) addresses the same problem, but it uses all possible state transitions feasible or not. The authors show that this can cause problems for reward functions that are technically equivalent in the feasible state transition state, but unequal on the infeasible transitions (which should not matter). The authors evaluate the method in "bouncing balls" (a discrete navigation task) and "reacher"  (robotic manipulation) environments. The results show that the proposed distance is more accurate compared to EPIC (distance is zero for "FEASIBILITY", 2 reward functions that are technically different but practically the same) and the distances are more aligned with the learned policies scores.

**Summary Of The Review:**

The paper proposes an incremental method that improves over past work for to ignore the infeasible state transitions. The idea is incremental, but has a good potential. The evaluation could be improved to show the impact of the given method. I think the paper can also use a higher level use-case of reward comparison where the proposed method is used to design a reward function to learn a problem that is otherwise unsuccessful.

---

> ### Author Response · Authors · 2021-11-16
> **Response to Reviewer vDTh**
>
> Thank you for reviewing the paper and providing feedback! We have responded below to individual quotes from your review.
>
> > The method assumes access to the dynamics model, and the authors state that this can be approximated (or learned) as well. I believe that an example based on learned dynamics model would make the paper much better.
> - We have attempted to address this comment by providing results for DARD when using a learned transition model in the environments considered in the main body of the paper. Please see section 5 for a discussion of the results, and section A.1.4 for details on model training.
> - While we believe more research is merited around this topic, particularly involving learned transition models in high-dimensional (e.g., visual) environments, we hope these additional results provide some insight into the performance of DARD when using learned transition models.
> - On a related note, the use of a *learned* transition model with DARD is an important special case of applying DARD with an *erroneous* transition model. We have added section A.3.1 in order to categorize and explain the types of errors DARD might make as a result of errors in the transition model.
>
> > Reading the paper, it was not easy for me to understand how the proposed method can be used in a large scale reward designing process. I think that the theme of "reward designing based on DARD for a very complex (or unknown) problem" could make the paper more impactful.
> - Thanks for raising this point! We have attempted to address this comment by adding a section in the appendix that describes two potential real-world applications of reward comparison metrics such as DARD. The first application considers the problem of benchmarking Inverse Reinforcement Learning algorithms. The second application considers the task of validating learned reward functions for deployment as part of an autonomous system. Please see section A.3.3 for details. We hope this section clarifies how DARD might be used in practical applications, and welcome feedback on it.
>
> > It would be very beneficial if the authors could provide a scenario where the DARD compares hundreds (or thousands) of arbitrarily chosen reward functions, and based on its results the authors could train a policy that outperforms the other.
> - Thank you for the suggestion. We are working on an experiment that compares a large number of reward functions, and evaluates the extent to which DARD is predictive of their associated policy return. We hope to complete this prior to the end of the review period.
>
> Please see our general response for a discussion of our view of the contribution of the research relative to prior work.

---

> > ### Author Response · Authors · 2021-11-18
> > **Response to Reviewer vDTh**
> >
> > We have made changes since our previous response addressing your feedback regarding the need for an experiment considering a large number of reward functions:
> >
> > > It would be very beneficial if the authors could provide a scenario where the DARD compares hundreds (or thousands) of arbitrarily chosen reward functions, and based on its results the authors could train a policy that outperforms the other.
> > - We have added an experiment to the paper that evaluates DARD in the context of a large number of randomly generated reward functions. For the experiment, we generate reward functions randomly for the Reacher environment, and then plot the DARD distance between those rewards and the ground-truth reward against the policy return of a policy optimized for each randomly generated reward. The results indicate that DARD maintains its predictiveness of policy return in this setting, though there were a number of interesting insights that resulted from performing the experiments (please see section A.2.3 in the revised paper for details). We hope this addresses your feedback, and welcome further comments regarding these additional experiments.

---

> > > ### Comment · Reviewer_vDTh · 2021-11-24
> > > **Thanks for the response**
> > >
> > > I would like to thank the authors for their response and their effort to address my comments. I believe that the changes they made are very helpful (at least for me) to find the paper more impactful. Although I would prefer the "use case for practical applcations" in the introduction rather than appendix, but it still helps me understand it better.
> > > I believe that the current version of the paper is better than I initially assessed, and can be accepted. So I'll raise my score accordingly.

---

> > > > ### Author Response · Authors · 2021-11-29
> > > > **Response to Reviewer vDTh**
> > > >
> > > > Thank you for following up and for the additional suggestion! We will add a brief (due to space constraints) description of the use cases as well as a reference to section A.3.3 to the introduction if the paper is accepted (because we can not submit further revisions at this point in the review process).

---

### Author Response · Authors · 2021-11-16
**General Response**

We would like to thank the reviewers for their helpful feedback and insightful comments.

**To summarize, the reviewers appear to agree on the following:**
1. The problem setting of comparing rewards without policy learning and evaluation is well-motivated.
2. The proposed method is technically sound.
3. Experiments indicate that the proposed method leads to improved performance over EPIC when considering learned reward functions.

We additionally thank the reviewers for their thorough constructive feedback. In response to this feedback, we would like to (i) briefly summarize the algorithmic contribution of DARD, (ii) describe additional experiments we have performed to address feedback, and (iii) describe additional discussions we have added to the paper in response to feedback.

**Summary of the algorithmic contribution of the paper:**
- The primary existing method of direct reward function comparison allows for avoiding policy evaluation, but suffers from poor performance when comparing learned reward functions, which are likely to be critical to many common use cases of reward comparisons.
- The key insight in DARD is in recognizing that reliable comparison of learned reward functions requires that these reward functions be evaluated on realizable transitions.
- Although operationalizing this insight builds on prior work, doing so produces an algorithm with significantly expanded utility: it enables reliable, direct comparison of learned reward functions without policy evaluation provided the transition dynamics of the system are approximately known (or learned with reasonable fidelity).

**Additional experiments**
- We have added experiments studying the performance of DARD when transition dynamics are learned for both of the simulation environments in the paper. Results suggest that even with learned dynamics, DARD maintains its predictiveness of policy returns and produces distance values similar to those resulting from the use of exact transition dynamics. Although the simulation environments are relatively low-dimensional (and as a result facilitate learning of high-fidelity transition models), this result suggests that DARD can be used effectively even when transition dynamics are not known apriori. Please see sections 5, A.1.4, A.3.1, and A.3.2 for results and detailed discussion.

- We intend to further expand on the experimental results as discussed in our responses to individual reviewers, time permitting.

**Additional discussion sections**
- We have added two discussion sections in response to feedback. The first (section A.3.1), discusses how errors in the transition model can impact the performance of DARD. The second (section A.3.3), discusses two potential applications of reward comparison metrics in order to clarify how DARD might be applied in real-world settings.

---

### Decision · Program_Chairs · 2022-01-20

**Decision:**

Accept (Spotlight)

**Comment:**

The paper proposes a new pseudometric, DARD, for comparing reward functions that avoid policy optimization. DARD builds on a recent work by Gleave et al. 2020 where the pseudometric EPIC was proposed. In contrast to EPIC, DARD operates on an approximate transition model and evaluates reward functions only on transitions close to their training distribution. Empirical experiments in different domains demonstrate the effectiveness of the proposed pseudometric. The reviewers acknowledged the importance of the studied problem setting and generally appreciated the results. I want to thank the authors for their detailed responses that helped in answering some of the reviewers' questions and increased their overall assessment of the paper. At the end of the discussion phase, there was a clear consensus that the paper should be accepted. The reviewers have provided detailed feedback in their reviews, and we strongly encourage the authors to incorporate this feedback when preparing a revised version of the paper.